



# A standardized index for assessing sub-monthly compound
# dry and hot conditions
Jun Li[1], Zhaoli Wang[1,2], Xushu Wu[1,2,*], Jakob Zscheischler[3,4], Shenglian Guo[5],

4                          Xiaohong Chen[6]

[1] *School of Civil Engineering and Transportation, State Key Laboratory of Subtropical*
*Building Science, South China University of Technology, Guangzhou 510641, China.*
[2] *Guangdong Engineering Technology Research Center of Safety and Greenization for*
*Water Conservancy Project, Guangzhou 510641, China.*
[3] *Climate and Environmental Physics, University of Bern, Sidlerstrasse 5, 3012 Bern,*
*Switzerland.*
[4] *Oeschger Centre for Climate Change Research, University of Bern, Bern, Switzerland.*
[5] *State Key Laboratory of Water Resources and Hydropower Engineering Science,*
*Wuhan University, Wuhan 430072, China.*
[6] *Center for Water Resource and Environment, Sun Yat-Sen University, Guangzhou*
*510275, China*
*∗Correspondence: xshwu@scut.edu.cn.*




**Abstract:** Compound dry and hot conditions pose large impacts on ecosystems and
society worldwide. A suite of indices are proposed for the assessments of droughts and
heatwaves previously, yet there is no index available for incorporating the joint
variability of dry and hot conditions at sub-monthly scale. Here, we introduce a daily-
scale index, termed as the standardized compound drought and heat index (SCDHI), to
measure the intensity of compound dry and hot conditions. SCDHI is based on the daily
drought index (the standardized antecedent precipitation evapotranspiration index
(SAPEI)) and the Standardized Temperature Index (STI) and a joint probability
distribution method. The new index is verified against real-world compound dry and
hot events and the related observed vegetation impacts in China. SCDHI can not only
monitor the long-term compound dry and hot events, but also capture such events at
sub-monthly scale and reflect the related vegetation activity impacts. The identified
compound events generally persisted for 25-35 days and the southern China suffered
from compound events most frequently. In future, the frequency, duration, severity and
intensity of compound events increase throughout China in response to anthropogenic
climate change, of which the frequency would increase by 1-3 times and the duration
and severity increase by 50%, independent of the emission scenarios. The new index
can provide a new tool to quantify sub-monthly characteristics of compound dry and
hot events, conducive to the timely monitoring of their initiation, development, and
decay which are vital for decision-makers and stake-holders to release early and timely
warnings.
**Keywords:** compound event; SCDHI; SAPEI; sub-monthly scale; China



## 1 Introduction

Compound dry and hot event (CDHE) have been observed for all continents in
recent decades (Hao et al., 2019; Mazdiyasni and AghaKouchak, 2015; Manning et al.,
2019; Sutanto et al., 2020). The frequent CDHEs have led to more devastating impacts
on natural ecosystems and human society than individual events (Zscheischler et al.,
2014; Chen et al., 2019; Hao et al., 2018a). For example, Russia was simultaneously
struck by an unprecedented drought and hot in the summer of 2010, which caused large-
scale crop failures, wildfires, and human mortality (Zscheischler et al., 2018).
Unfortunately, the extreme droughts and hots are expected to occur more frequently in
the coming decades under global warming, which potentially results in more compound
events in many parts of the world, especially for wet and humid regions (Wu et al.,
2020; Swain et al., 2018, Zscheischler and Seneviratne, 2017a). Therefore,
understanding such events are of crucial importance to provide the most fundamental
information to help disaster mitigation.
Much effort has been made to study the compound events in recent years. Utilizing
different thresholds to define the concurrent climate extremes for a specific period, the
frequency of compound events has received  a great deal of attention (Wu et al., 2019;
Zhang et al., 2019). Although this approach can detect compound event occurrence, it
fails to quantitatively measure compound event characteristics such as duration,
severity, and intensity, and is inconvenient for comparison of compound event
characteristics through different climates (Wu et al., 2020). Therefore, to overcome
these shortages, several joint climate extreme indices have been proposed for analyzing
the characteristics of the compound events. For example, the climate extreme index
integrated by temperature and soil moisture extremes was presented for monitoring
trends in multiple types of climate extremes across large regions, and has been





employed to assess changes in spatial extent (Gallant et al., 2014). In recent years,
several compound dry and hot indices have been developed. For example, the
Standardized Dry and Hot Index based on the ratio of the marginal probability
distribution functions of precipitation and temperature was proposed to measure the
extreme degree of a compound drought and hot extreme event (Hao et al., 2018). Hao
et al. (2019) recently proposed the Standardized Compound Event Indicator (SCEI) to
assess the severity of compound dry and hot events by jointing the marginal distribution
of Standardized Precipitation Index (SPI) and Standardized Temperature Index (STI)
using the copula theory. These two joint indices provide useful tools to improve our
understanding of the frequency, spatial extent and severity of CDHE. However, they
are inevitably subjected to some shortcomings including the fixed monthly scale and
the disregard of evapotranspiration, which may limit their use in monitoring the detailed
evolution of compound dry and hot events.

With the occurrence of extreme climate (e.g. high temperature, low humidity, and

sunny skies), droughts can evolve rapidly (Chen et al., 2019; Koster et al., 2019; Mo
and Lettenmaier, 2015; Otkin et al., 2018; Yuan et al., 2019; Li et al., 2020a). Such
extreme weather can appear within a short period without resulting in long-lasting
compound events, but rather, short-term droughts and heatwaves lasting a few weeks
or even days (Mo and Lettenmaier, 2016; Zhang et al., 2017). Severe concurrent
drought and heat can suddenly strike a region with a relatively short duration when
extreme weather anomalies persist over the same region (Röthlisberger and Martius,
2019; Wang et al., 2016). Concurrent short-term drought and hot can pose greater
potential socio-economic risks because the combination of these events can exacerbate
their respective environmental and societal impacts (Kirono et al., 2017; Schumacher
et al., 2019; Sedlmeier et al., 2018). Specifically, even short-term concurrent dry and





hot extremes can lead to significant agricultural loss if they occur within sensitive stages
in crop development such as emergence, pollination, and grain filling (Zhang et al.,
2019). For example, a strong precipitation deficit along with record high temperatures
have led to severe impacts during May and early June in 2012 across the central U.S.
(Ford and Labosier, 2017; Otkin et al., 2013). Such short-term concurrent dry and hot
events regularly inflict widespread agricultural crop losses and drastically cut down
livestock population, making it one of the most costly natural hazards in the U.S. history
at tens of billions of economic losses (Anderson et al., 2016; Otkin et al., 2019). Under
climate change, short-term concurrent dry and hot extremes are expected to increase
(especially for humid regions), potentially causing substantial damage to natural
ecosystems and society (Li et al., 2020b; Sun et al., 2019). To improve understanding
of such short-term compound events and make early and timely warnings, decision-
makers and stakeholders require more detailed information such as the start time,
severity, and the projected tendency in the coming days rather than the average state at
a fixed monthly scale. Correspondingly, sub-monthly scale indices for characterizing
short-term compound dry and hot events are needed. Through the influence of
evapotranspiration, short-term meteorological variables (e.g., solar radiation and
sunshine duration) are considered an important factor in drought and heatwave
concurrences (James et al., 2010). For example, the largely increase in sunshine
duration due to clear sky creates excessive evapotranspiration, which in turn decreases
soil moisture (Ford et al., 2015). More surface sensible heat fluxes is transferred to the
near-surface atmosphere to further increase air temperatures and prohibit precipitation
(Miralles et al., 2019; Vogel et al., 2018). Together, these land-atmosphere interactions
create favorable conditions for concurrent drought and heatwaves (Mo and Lettenmaier,
2016; Otkin et al., 2018). Thus, the development of a compound drought and heat index





should consider other important drought/hot-related factors including temperature and
precipitation (e.g. evapotranspiration).
The complexity of compound events makes it an unusual task to develop a simple
and robust index to quantify their past and future changes (Zscheischler et al., 2020). A
suite of indices are proposed for the assessments of droughts and heatwaves previously,
yet there is no index available for incorporating the joint variability of dry and hot
conditions at sub-monthly scale. Here we aim to formulate a compound drought and
heat index, called the standardized compound drought and heat index (SCDHI), for
monitoring and analyzing compound dry and hot events at sub-monthly scale. To
achieve this aim, we combine a daily scale drought index, the standardized antecedent
precipitation evapotranspiration index (SAPEI), which simultaneously considers
precipitation and potential evapotranspiration (PET), with a daily scale standardized
temperature index (STI). We investigate the characteristics such as frequency, duration,
severity, and intensity of CDHEs during the historical (1961-2018) period and project
their changes in China for the future (2050-2100) under different emission scenarios.
This index can provide a new tool to quantify the characteristics of CDHEs, and can
monitor the CDHE at multiple time scale (e.g., daily, weekly and monthly) to provide
detailed information on their initiation, development, decay, and trends.
**2 Methods**
**2.1 data**
Daily meteorological datasets covering 1961 to 2018 were collected from 2239
observational stations across the non-arid region in China (Fig. 1), which include
precipitation (P), maximum air temperature ($T_{max}$), mean air temperature ($T_{men}$),
minimum air temperature ($T_{min}$), relatively humidity (RH), wind speed (WS), and
sunshine duration. All of these meteorological data with strict quality control are





available from the China Meteorological Administration (http://cdc.nmic.cn/home.do)
and the Resources and Environmental Science Data Center, Chinese Academy of
Sciences (http://www.resdc.cn/Default.aspx). The kriging method was applied to
interpolate these observational station data into $0.25 \times 0.25°$ gridded data.
The two commonly used indices (i.e., monthly Palmer Drought Severity Index
(PDSI) and Standardized Precipitation Evapotranspiration Index (SPEI) were employed
for comparison. PDSI and SPEI were computed from the same meteorological data
described above. The conventional PDSI was empirically derived using the
meteorological data of the central USA with its semi-arid climate. The portability of
the conventional PDSI is thus relatively poor (Liu et al., 2017). In this study, PDSI was
calculated according to the China national standard of classification of meteorological
drought with standard number of GB/T 20481-2017. The PDSI calculation procedure
of this standard was built based on long-term meteorological data of in-situ stations
evenly distributed around China, hence well monitor drought in China (Zhong et al.,
2019a). The 0.25°-daily root zone (0 - 100 cm) soil moisture dataset obtained from
Community Land Model (CLM) of the Global Land Data Assimilation System
(GLDAS) was also used in this study. The dataset from 1961 to 2014 were downloaded
from the Goddard Earth Sciences Data and Information Services Center (Rodell et al.,
2004). The GLDAS CLM soil moisture dataset captures dry and wet conditions in
China well (Bi et al., 2016; Feng et al., 2016). In addition, 8-day leaf area index (LAI)
of the MOD15A2H from 2003 to 2018 were collected. These data were resampled to
0.25° spatial resolution, and then the Z-score was used to calculate the LAI anomalies.
We further used eight global climate models from the Coupled Model
Intercomparison Project Phase (https://esgf.llnl.gov/) (Taylor et al., 2012), including
CanESM2, CNRM-CM5, CSIRO-Mk3.6, MIROC-ESM, MPI-ESM-LR, BCC-CSM1-



1, IPSL-CM5A-LR, and MRI-CGCM3, were used to project the future climate
conditions. These GCMs exhibit good performance to simulate the key features of
precipitation and temperature in China (Jiang et al., 2016; Yang et al., 2019). We
obtained daily climate variables (i.e., P, $T_{max}$, $T_{min}$, $T_{men}$, WS, RH, and shortwave and
longwave radiations) for the historical (1961-2005) and future (2030-2100) periods for
the three Representative Concentration Pathways (RCPs) including RCP 2.6 (low
emission scenario), RCP 4.5 (moderate emission scenario) and RCP 8.5 (high emission
scenario). All of the GCM outputs were based on the first ensemble member of each
model, referred to as *r1i1p1* in all of the experiments. The detailed information on these
GCMs is shown in Table S1.

## 2.2 Development of SCDHI

The SCDHI is a compound drought and heat index based on a daily drought index
and the Standardized Temperature Index (STI), which is computed in a similar fashion
as the Standardized Precipitation Index (Zscheischler et al., 2014). The calculation of
daily STI is similar to monthly STI, but for standardizing daily temperature. For
example, with respect to one certain grid point, the 1 January STI are computed on the
1 January temperature datasets observed during 1961-2018 at each grid point. We firstly
formulated a daily scale drought index, i.e. the standardized antecedent precipitation
evapotranspiration index (SAPEI), by considering both precipitation and PET.
Afterward, the joint distribution method was employed to compute the SCDHI.

### 2.2.1 Formulation of daily-scale drought index

Li et al. (2020b) have proposed the daily-scale drought index (SAPEI) that
considers both precipitation and PET. However, the primary limitation of this index is
that it has a fixed temporal scale and cannot reflect the dry and wet condition at different
time scales. Hence, in this study, we developed the multiple time scale (i.e., 3-, 6-, 9-,



and 12-month) daily drought index. Here, we followed the same nomenclature proposed
by Li et al. (2020b) to refer to a daily standardized drought index (SAPEI) based on
precipitation and PET. SAPEI is simple to calculate, and uses the antecedent
accumulative differences between precipitation and PET to represent the dry and wet
condition of the current day. The calculation procedure is described below.

The Penman-Monteith method (Allen et al., 1998) was firstly used to compute PET.

With a value for PET, the daily difference between precipitation and PET was
calculated to reveal climatic water balance (precipitation minus PET). To reflect dry
and wet conditions of the day, the antecedent water surplus or deficit ( $D$ ) was
calculated through the following equations:

$$D = \sum_{i=1}^{n} \left( p - PET \right)_{i} \tag{1}$$

Where $n$ is the number of previous days.

The $D$ values can be aggregated at different time scales, such as 3, 6, 9 months,

and so on. A probability distribution was used to fit the daily time series of $D$. Given
that different probability distributions may cause differences in drought indices (Stagge
et al., 2015), to select the most suitable distribution, several commonly probability
distributions including the general extreme value, log-logistic, lognormal, Pearson III,
generalized Pareto, exponential, and normal distributions, should be used to fit the $D$
series. In the study of Li et al. (2020b), Shapiro-Wilk and Kolmogorov-Smirnov (KS)
test have been used applied for optimal probability distribution selection by comparing
the empirical probability distribution with a candidate theoretical probability
distribution.  They suggested that the log-logistic distribution is more suitable for
SAPEI. Moreover, previous researches have demonstrated that the log-logistic
distribution is suitable for standardizing drought indices, e.g. SPEI (Vicente‐Serrano et



al., 2010). Therefore, we chose the log-logistic distribution to compute SAPEI. Once
the daily $D$ series were fit to a probability distribution, cumulative probabilities of the
$D$ series were obtained and transformed to standardized units (SAPEI) using the
classical approach of Barton et al. (1965).
**2.2.2 Construction of SCDHI**

The SCDHI was established through copula theory, which can combine the

candidate variables into one numerical expression. This approach not only realizes a
projection from multiple dimensions to a single dimension, but also the marginal
distributions of the candidate variables combined with their original structures can be
fully preserved within the constructed joint distribution. Hence, the copula-based index
provides an objective description of the compound events (Hao et al., 2018b; Terzi et
al., 2019).

There are many copula families available, which have widely been used for joint

bivariate distributions (Terzi et al., 2019; Zhang et al., 2018). Among then, Clayton,
Gumbel, Normal, T, and Frank copula perform well for jointing bivariate
hydrometeorological variables (Ayantobo et al., 2018; Liu et al., 2019), and thus were
employed to establish the bivariate joint probability distribution in this study. Assuming,
the two random Gaussian variables $X$ and $Y$ representing SAPEI and STI,
respectively, the CDHE can be identified as one variable $X$ lower than or equal to a
threshold $x$, and the other variable $Y$ higher than a threshold $y$ at the same time. The
joint probability $p$ of the CDHE can then be expressed as:

$$p = P(X \le x, Y \ge y) = u - c(u,v) \qquad (2)$$

where $u$ was the $X$ marginal distribution, and $c(u,v)$ was the joint probability





distribution.
This joint cumulative probability $P$ could be treated as an indicator, where smaller
$P$ values denote more severe condition of CDHE. However, $P$ to the given marginal
sets, $P$ values in different seasons or areas reflected different conditions and are thus
not comparable. Hence, the joint probability $P$ was transformed to a uniform
distribution by fitting a distribution $F$, which was then standardized as an indicator to
characterize CDHEs. Once the $P$ series at each day were fitted to a copula, the $P$
series were transformed to standardized units. SCDHI can be estimated by taking the
inverse of joint cumulative probability (p) as:

$$SCDHI = \varphi^{-1}(F(P(X \le x, Y \ge y)))  \qquad (3)$$

where $\varphi$ is the standard normal distribution function. the distribution $F$ was
estimated based on the Yeo-Johnson transformation formula (Yeo and Johnson, 2000).
Following the categories of compound dry and hot conditions as suggested by (Wu
et al., 2020), we defined five categories of compound dry and hot conditions, including
abnormal, light, moderate, heavy and extreme compound drought-hot, as shown in
Table 1.
We used Akaike information criterion (AIC), Bayesian information Criterion (BIC),
and KS statistics as goodness-of-fit measures to select an appropriate copula. These
statistical measures have been commonly used for estimating the goodness of fit of a
proposed cumulative distribution function to a given empirical distribution function
(Liu et al., 2019; Terzi et al., 2019). The AIC, BIC, and KS statistics are presented in
Fig. S1-3. According to the evaluation metrics, there was a good agreement between
the empirical and parametric copulas. Particularly, the performance of Frank copula
slightly outweighed those of the other three copulas. Therefore, the Frank copula was
utilized to establish the joint probability function and construct SCDHI in this study.
Note that the SCDHI under three future scenarios is also used the Frank copula, while
the parameters are assessed by future scenarios data. The SCDHI development was
illustrated in Fig. S4.

Furthermore, to verify the ability of SCDHI to capture the compound dry and hot

event, three verification metrics were used (i.e., probability of detection (POD), false
alarm ratio (FAR), and critical success index (CSI)) (Zhang et al., 2018).

$$POD = H / (H + M) \qquad (4)$$

$$FAR = F / (H + F) \qquad (5)$$

$$CSI = H / (H + F + M) \qquad (6)$$

where $H$ (Hit, observed drought-hot) refers to the number of grids when SAPEI

and STI is subjected to grade 1 (G1) - grade 4 (G4) and SCDHI is subjected to G1-G4;
$M$ (Miss) denotes the number of grids when SAPEI and STI is between G1 to G4 and
SCDHI is subjected to other grades than G1-G4; $F$ (false alarm) denotes the number
of grids when SAPEI and STI is subjected to other grades than G1-G4 but SCDHI is
subjected to grades of G1-G4.
**3 Results and Discussion**
**3.1 Evaluation of SAPEI**

The SCDHI was established based on the STI and daily-scale drought index, i.e.,

SAPEI. However, no previous studies have tested the (daily) drought monitoring
performance of SAPEI. When developing a drought index, rigorous testing is required
with respect to its applicability before it is applied in drought monitoring. Fig. 2 shows
the spatial distributions of the correlations between SAPEI and SPEI/PDSI/soil
moisture across China. The monthly mean SAPEI at 3-, 6-, 9- and 12-month scale all
showed strong agreement with the SPEI in China, with correlation coefficients higher
than 0.8 (p < 0.01), indicating that the monthly SAPEI at multiple time scale calculated
from the daily value could have the same capability of monthly drought monitoring as
SPEI. The 3-, 6-, 9- and 12-month SAPEI generally showed good correlation with PDSI,
and 3-month SAPEI and PDSI generally correlate closely, with correlation coefficients
higher than 0.6 (p < 0.01), while the 12-month SAPEI displayed weak correlation with
PDSI in south China. For daily SAPEI at 3-month scale and soil moisture, a close
correlation was detected in south China, while relatively weak correlation is found in
north China. The correlation between SAPEI and soil moisture increased in magnitude
and spatial extent at time scales of 6-12 months. For 12-month SAPEI, correlation
coefficient was generally greater than 0.6 for a majority of China. This phenomenon
implied that the short-time scale SAPEI was more sensitive to precipitation change, and
thus could be more suitable for meteorological drought, while the long-time scale (more
than five month) SAPEI was more closely related to soil moisture and can be applied
for agricultural drought monitoring. Overall, these analyses indicate that the SAPEI at
daily and monthly scale showed reliability in drought monitoring

To further test the drought monitoring performance of the SAPEI, typical drought

events were chosen as case studies. During recent decades, several well-known large-
scale drought events have hit China, including the droughts in winter of 2009 to spring
of 2010, and in 2011 (Lu et al., 2014; Yu et al., 2019). In this study, the drought regimes
during these events were taken as case studies to evaluate the drought monitoring
performance of SAPEI at 3- and 6-month time scales (Sun and Yang, 2012). We firstly





showed the monthly evolution of these events by the monthly mean SAPEI, SPEI, and
PDSI, and then analyzed the daily evolution of drought in space and time in the most
affected areas according to SAPEI and soil moisture.

**3.2.1 Drought events during 2009-2010**

Fig. S5 illustrates the monthly changes in the 2009/10 drought monitored by the
PDSI, SPEI, and SAPEI at 3- and 6-month scale. This drought started to appear in most
of China (except for the central and northeast China) in September 2009, and then
persisted in most of China during October to December 2009; during this period,
drought conditions became more severe in south China. The drought in north and east
China gradually faded away during January and March in 2010. In contrast, in
southwest China (SWC) the drought intensity became rather strong during the same
period. The severe dry condition continued in SWC during April in 2010, while drought
in the rest of China gradually disappeared in this period. After that, dry conditions in
SWC gradually relieved from May to June in 2010, but did not disappear. The monthly
drought evolution based on SAPEI was generally similar with that of SPEI and PDSI.
Despite being located in the humid climate zone, SWC suffered from exceptional
drought during the autumn of 2009 to the spring of 2010 (Lin et al., 2015). During this
drought, more than 16 million people and 11 million livestock faced drinking water
shortages, with direct economic losses estimated at 19 billion yuan in SWC (Lin et al.,
2015). We selected this event in SWC as the first case study, and reveal detailed
spatial and temporal change of this event at daily scale based on SAPEI and soil
moisture (Fig. 3 and 4). During September 1 to 30 of 2009, the drought started to appear
in the region, and dry conditions became worse and spread throughout nearly the entire
SWC from October 1 to November 15 of 2009. Severe dry conditions then stayed in
the region for 152 days from November 15 to April 15 of 2010, with high intensity.





Afterwards, severe drought was gradually relieved from April 15 to June 15. The
drought diminished over time in most parts of southwest China by the end of June.

**329    3.2.2 Drought events in 2011**

The monthly changes in the 2011 drought is illustrated in Fig. S6. The drought
pattern monitored by SAPEI generally shows a good agreement with those by SPEI and
PDSI. More specifically, the drought mainly started in north China in January, while in
March it spread to most of China, and drought conditions in lower reaches of the
Yangtze River basin became serious. In April to May, severe dry conditions persisted
in the middle and lower reaches of the Yangtze River Basin (MLR-YRB), and extended
from the YRB to southern China. In August, the drought moved westward and reached
the edge of southwestern China. Severe drought persisted in the region during
September and October, but it gradually faded away in November and December. The
results monitored by the SAPEI are generally consisted with the findings of Lu et al.

(2014).

The 2011 drought event was particularly unusual in the LR-YRB. The MLR-YRB
is generally in a wet condition, nevertheless, suffered its worst drought in the 50 years
during the spring. The severe drought caused shortage of drinking water for 4.2 million
people. 3.7 million hectares of crops were damaged or destroyed. Moreover, the heavy
drought led to more than 1,300 lakes devoid of all water in Hubei province (Xu et al.,
2015). The temporal and spatial evolution of this event in MLR-YRB described by daily
SAPEI and soil moisture was shown in Fig. 5-6. The drought started to appear in the
north part of the MLR-YRB in early February of 2011, and then gradually expanded to
the whole MLR-YRB during early February and March 15. The severe drought
condition persisted in this region for 78 days (from March 15 to May 31). Afterwards,
there was a tendency toward alleviating drought conditions, and most of MLR-YRB





was under light and moderate drought conditions.
The previous detailed analysis showed that the SAPEI not only captures monthly
characteristics of droughts, but also has the potential to track droughts at sub-monthly
scale. Though the input data (including precipitation and PET) of SAPEI are similar to
SPEI, the rationale of the index is different from SPEI. It was calculated for each day
and considers the water surplus or deficit of that day and the previous days. SPEI was
commonly employed to monitor and analyze the monthly or longer-scale droughts
(Vicente-Serrano et al., 2010). It thus may not be appropriate to apply the SPEI at
shorter timescales (e.g., daily or weekly), because of the inherent problem in the
construction of the index. Although SPEI gives a full and equal consideration to the
water surplus or deficit in the period of the considered time scale, it does not consider
the water surplus or deficit in the days before the period. If the scale is very short, this
may cause problems. For a 7-day period, for example, if there is no precipitation during
the period, it may be regarded as a drought period when compared with historical
records (the method used by the SPEI); however, if there is a heavy precipitation just
before the period, then the 7-day period probably remains wet and is unlikely to
experience drought condition during such a short time. Previous studies have
demonstrated the disadvantage of SPEI for short-time scale drought monitoring (Lu,
2009; Lu et al., 2014; Li et al., 2020b).
In summary, the SAPEI meets the requirements of a drought index, given the fact
that it shows reliable and robust ability for drought analysis and monitoring. Like the
SPEI, SAPEI includes multiple time scales (3-, 6-, 9-, and 12- month) to monitor
droughts at monthly resolution.  However, SAPEI has the advantage over SPEI
regarding sub-monthly drought monitoring.





## 3.2 Evaluation of SCDHI


The SCDHI was developed by joining the marginal distribution of the SAPEI and
STI. Though the copula method has been widely utilized to connect bivariate
distribution, the property of SCDHI in capturing CDHEs still needs to be tested. Fig. 7
shows the spatial distributions of the correlations between SCDHI and SAPEI/STI at
daily scale across China. The SCDHI all showed strong ($p < 0.01$) correlation with the
SAPEI at 3-, 6-, 9- and 12-month scale in China, with correlation coefficients higher
than 0.7. A significant correlation ($p < 0.01$) was also detected between STI and SCDHI
at multiple scales. Fig. 8 shows the spatial pattern in POD, FAR, and CSI when the
drought and hot events observed by SAPEI and STI, respectively, were related to
compound drought-hot event detected by SCDHI at 3-, 6-, 9- and 12-monthly scale. As
shown in Fig. 8, POD is close to 1 and FAR is close to 0, implying that SCDHI can
well detect in most of the areas where the droughts and hots were detected by SAPEI
and STI. The values of CSI indicated that the ratios of drought-hot affected areas
detected by SAPEI and STI to the drought and hot areas detected by SCDHI were close
to one. Overall, these analyses implied that SCDHI can well monitor droughts and hots
that can be successfully captured by SAPEI and STI. The SCDHI thus detects CDHEs
that are identified separately by the coincidence of low SAPEI and high STI. In addition,
the SCDHI detects events that are very extreme in either the SAPEI or the STI and
moderate in the other variable but thus still cause substantial damage (Zscheischler et
al., 2017b). Furthermore, the SCDHI is able to quantify the magnitude of CDHEs.
To further test the drought-heat monitoring performance of the SCDHI, two typical
CDHEs were chosen as case studies according to the Yearbook of Meteorological
Disasters in China. One was a well-known compound drought and heatwave striking
Sichuan-Chongqing region (SCR) with serious consequences during summer of 2006





(Wu et al., 2020), and the other occurred in southern China with adverse impacts on
agriculture during July to September of 2009 (Wang et al., 2010). SCR experienced
continuous extreme temperature during mid-June to late August 2006. The duration and
severity of this hot event were the worst on the historical record. Simultaneously, a
heavy drought occurring once in 100 years hit this region. During this compound event,
a population of over ten million was confronted with drinking water shortage, about
twenty thousand $km^2$ of cropland suffered serious losses, and more than one hundred
times forest fire broke out. Local governments issued the most serious arid warning
(Zhang et al., 2008). Thus, we take this typical drought-hot event as first case studies
to evaluate the drought/hot monitoring performance of SCDHI. The monthly spatial
pattern of this compound event in SCR is shown in Fig. S7, indicating that SCR during
summer in 2006 experienced the moderate to extreme compound dry and hot conditions.
Fig. 9 maps the spatial pattern of this compound event and its impact on vegetation
from mid-June to late August. This event started to appear in SCR in mid-June 2006,
and gradually spread throughout the whole SCR during June 19 to 26. The moderate
dry-hot condition then persisted in the entire SCR from June 27 to August 5 in 2006,
lasting for 40 days. The negative LAI was scattered in some of the dry-hot affected
areas. However, during August 6 to 21, the drought-hot event became even more
severe with the onset of extremely hot temperatures, causing negative vegetation
anomalies in most of the affected areas.
The monthly spatial pattern of another compound event in southern China during
July to September of 2009 is shown in Fig. S8. Overall moderate to heavy compound
dry and hot conditions are observed at monthly scale in this region. However, this event
showed large fluctuation at weekly scale. According to the Yearbook, the hot event was
divided into two periods: the first stage was from early to late July, and the other stage





was from mid-August to early September. The fluctuating compound event caused
adverse impact of crop pollination and grain filling, resulting in decrease of crop
production. Fig. 10 maps the spatial pattern of this event and its impact on LAI. In the
first stage, the drought-hot event hit the most of southern China during July 5 to 12, and
then it became severe in the west part of southern China during July 13 to 20. However,
the hot event suddenly disappeared from July 21 to 28, leading to disappearance of the
compound event in most of southern China (Fig. 10a). Afterward, the compound event
hit this region again from August 6 to 13, and its intensity was strong during August 14
to 21, with severe hot conditions. Subsequently, the intensity and spatial extent of the
compound event faded away in north of southern China during August 22 to 29. This
event extended to most of this region again from August 30 to September 14, with
severe dry and hot condition. The compound events still stayed in this region from
September 15 to 22 (Fig. 10b). Despite the short-term event, the anormal change in
vegetation was found in most of the dry-hot affected areas. This complex event
indicates that monthly analyses of the event can provide an overall situation, but is
not be able to capture the serious dry and hot conditions caused by a short-term extreme
climate anomaly at shorter time scales. Though such short-term compound event only
lasted for days or weeks, they lead to large agricultural losses if they occur within
sensitive stages in crop development (i.e., pollination and grain filling) (Mazdiyasni
and AghaKouchak, 2015). To provide timely information of the CDHEs, short-time
scale analyses and monitoring of such events are essential.

Overall, the changes in these two CDHEs based on SCDHI are consistent with the

national weather reports (http://www.weather.com.cn/). In summary, the SCDHI is able
to robustly and reliably capture CDHEs at sub-monthly scale, and potentially provide a
new tool to objectively and quantitatively analyze and monitor the characteristics of





CDHEs in time and space.

## 3.3 Application

Here, we evaluate and compare the spatiotemporal variation of characteristics of
CDHEs in China. More precisely, the CDHEs during growing season (April-September)
from 1961 to 2018 were identified based on 3-month scale SCDHI and run theory (Wu
et al., 2018), after which the frequency, duration, severity, and intensity of these events
were analyzed (A specific case to identify CDHE is shown in Fig. S9). We then
projected their future characteristics changes under the RCP 2.6, 4.5 and 8.5 from 2050
to 2100. Given that short-term concurrent dry and hot events    generally persist for at
least weeks (Otkin et al., 2018), only the events lasting for more than two weeks were
considered in this study.
Fig. 11 shows spatial patterns of characteristics of the CDHEs. A high frequency of
compound events was detected in southern China, with occurrence of every two years
on average, in contrast, the eastern Tibet Plateau and northeast China experienced fewer
compound events (Fig. 11a), which was generally consistent with the previous studies
(Liu et al., 2020; Wang et al., 2016). The CDHE generally lasted for about twenty-five
to thirty-five days in most of China, while in east Tibet Plateau, the CDHE persisted
for less than twenty days (Fig. 11b). The severity and intensity of the CDHE presented
relatively similar patterns and showed that most of eastern China experienced high
severity and intensity (Fig. 11 c-d). Overall, southern China suffered more frequent
CDHEs, with higher severity and intensity. Southern China is a humid region where
evapotranspiration is mainly controlled by energy supply because soil moisture is
usually sufficient. For given adequate soil moisture in the initiation of drought,
evaporative demand can increase rapidly during a short period when strong, transient
meteorological changes (such as extreme temperature) occur, which in turn exhaust soil





moisture to intensify drought conditions (Zhang et al., 2019, Otkin et al, 2018).
Moreover, vegetation over south China is usually abundant and plants tend to suck more
water from the soil when high temperatures occur, causing evapotranspiration increase
and soil moisture decline (Li et al., 2020c; Wang et al., 2016). More surface sensible
heat fluxes are thus transferred to the near-surface atmosphere to further increase air
temperatures (Mo and Lettenmaier, 2015). These land-atmosphere interactions
altogether cause the Bowen ratio to increase (Otkin et al., 2013, 2018), creating a
favorable condition for short-term concurrence droughts and hots. Therefore, CDHEs
are more likely to occur in humid regions with higher severity and intensity.

Fig. 12 illustrates the spatial patterns of change in frequency, duration, severity, and

intensity of the CDHEs under RCP 2.6, 4.5, and 8.5 scenarios. According to Fig. 12a,
the future (2050-2100) CDHE frequency under three scenarios in most of east China
will increase by about one to three times with respect to the reference period (1961-
2018). Under RCP 8.5 scenario, CDHE at about 4% of the study region is expected to
markedly increase by more than five times, which are scattered in the central to west
parts of China. The duration of CDHE across the east of the study region will mainly
show an increase of about 0.5 times, while duration in mid-west China potentially
increases by approximately 1.5 times under RCP 8.5 scenarios (Fig. 12b). The spatial
pattern of future severity change is similar to the duration; severity in most of east China
is projected to increase by about 0.5 time under three scenarios; however, CDHE
severity over mid-west China is expected to more than triple under RCP 8.5 (Fig. 12c).
The CDHE intensity in most of the study region exhibits slight increase for all scenarios
in comparison to the historical period.

The cumulative density functions (CDFs) of the CHDE frequency, duration,

severity, and intensity in historical and future periods were quantified, and the result is





shown in Fig. 13. A substantial change in the values of CHDE frequency, duration,
severity, and intensity was detected between the historical and future projections. The
frequency, duration, severity, and intensity of CHDEs will intensify throughout the
China in future scenarios compared to the historical reference, as marked by the
movement towards the right side of the CDF curves. Specifically, the cumulative
probability of CDHE frequency is expected to increase by more than 80% under three
scenarios, compared with the 95[th] percentile value in historical period (Fig. 13a). The
cumulative probability of duration would increase by about 72% under RCP 8.5
scenario, while increment under RCP 2.6 scenario is relatively small (17%), in
comparison to the 95[th] percentile in reference period (Fig. 13b). The severity cumulative
probability project to increase by 42% and 53% under RCP 2.6 and 4.5 scenarios
respectively, but even increase by 88% under RCP 8.5 scenario (Fig. 13c). An increase
of at least 42% is observed in the intensity cumulative probability, compared with the
n reference period (Fig. 13d). Such an increase in the frequency, duration, severity, and
intensity of CDHEs across China could be a new normal in future.
Global warming is very likely to exacerbate the prevalence of the CDHEs
(Pfleiderer et al., 2019). Trends are often present in individual variables (e.g.,
temperature, and precipitation), while can also occur in the dependence between drivers
of compound events, which consequently affects associated risks. The (negative)
correlation between seasonal mean summer temperature and precipitation is projected
to intensify in many land regions, leading to more frequent extremely dry and hot
conditions (Kirono et al., 2017; Zscheischler and Seneviratne, 2017a). Overall, the
frequency, severity, duration, and intensity of the CDHEs in China under global
warming will increase significantly. Effective measures need to be implemented to
decrease the $CO_2$ emissions for compound dry and hot event mitigation.



## 4 Conclusions


Under global warming, the compound dry and hot event tends to more frequent and
short-lived (i.e., days or weeks). Correspondingly, a compound drought and heat index
should be able to monitor such event at sub-monthly scales in order to timely reflect
dry and hot condition evolution. In this study, we developed a multiple time scale (e.g.,
3-, 6-, 9, and 12- month) compound drought and heat index, termed as SCDHI, to
monitor short-time (e.g., days or weeks) and long-time (e.g., months) compound event.
This index was established based on the daily drought index (i.e., SAPEI) and
Standardized Temperature Index (STI) using a joint probability distribution method.
Using the SCDHI, we then quantitively investigated the characteristics (i.e., frequency,
intensity, severity, and duration) of the CDHEs in China in historical period (1961-
2018), and revealed how they would change in the future (2050-2100) under
representative concentration pathway (RCP) 2.6, 4.5, and 8.5 scenarios. The main
conclusions of this study are presented as follows: The SCDHI can well monitor
simultaneous dries and hots detected by SAPEI and STI. The monthly SCDHI can
provide an overall situation of the compound dry and hot conditions, but sub-monthly
SCDHI can well capture fluctuation of simultaneous dries and hots within a month. It
also can reflect the impact of the compound dry and hot event on vegetation anomalies.
The SCDHI can offer a new tool to quantitatively measure the characteristics of the
CDHEs. It also can provide detailed information such as the initiation, development,
decay, and tendency of the compound event for decision-makers and stakeholders to
make early and timely warning. In the case study of the China, the southern China
suffered more frequent the CDHE, with higher severity and intensity. The CDHE
mainly lasted for twenty-five to thirty-five days in China. The frequency, duration,
severity, and intensity of compound events will intensify throughout the China in future.



The frequency will increase by about one to three times with respect to the reference
period. A region with fewer compound event (< 5) would exhibit a multi-fold (more
than five times) increase in the future. The duration across east areas mainly increased
by 0.5 times, while severity project to increase by about 0.5 to 1 times.

**Data availability.** The observed meteorological datasets are available at
http://cdc.nmic.cn/home.do. The CMIP5 datasets are available at https://esgf.llnl.gov.

**Author Contributions.** Conceived and designed the experiments: JL, WS. Performed
the experiments: JL, WS. Analyzed the data: JL. Wrote and edited the paper: JL, WS,
WZ, ZJ, GS, CX.

**Competing interests.** The authors declare that they have no conflict of interest.

**Acknowledgement**

The research is financially supported by the National Natural Science Foundation

of China (51879107, 51709117), the Guangdong Basic and Applied Basic Research
Foundation (2019A1515111144), and the Water Resource Science and Technology
Innovation Program of Guangdong Province (2020-29).











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





**Table**
Table 1 Categories of compound dry and hot conditions based on SCDHI.

| Category | Dry and hot condition | SCDHI |
|---|---|---|
| $G_0$ | Abnormal | (-0.80, -0.50] |
| $G_1$ | Light | (-1.30, -0.80] |
| $G_2$ | Moderate | (-1.60, -1.30] |
| $G_3$ | Heavy | (-2.0, -1.60] |
| $G_4$ | Extreme | $\leq$ -2 |






**Figure**


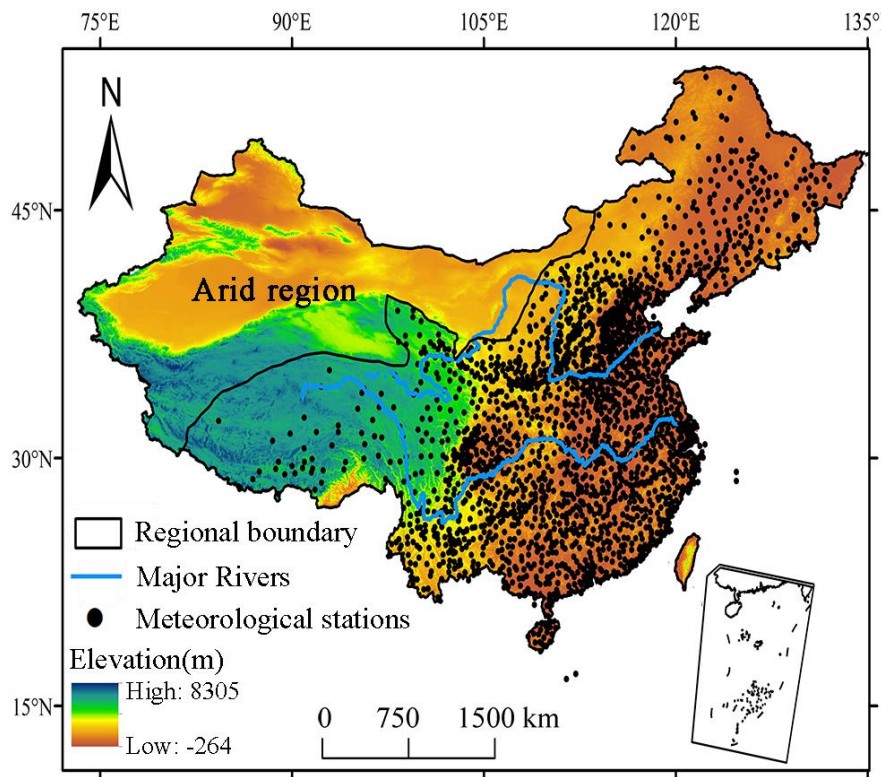


Figure 1 Geographical position of China and local of meteorological stations.








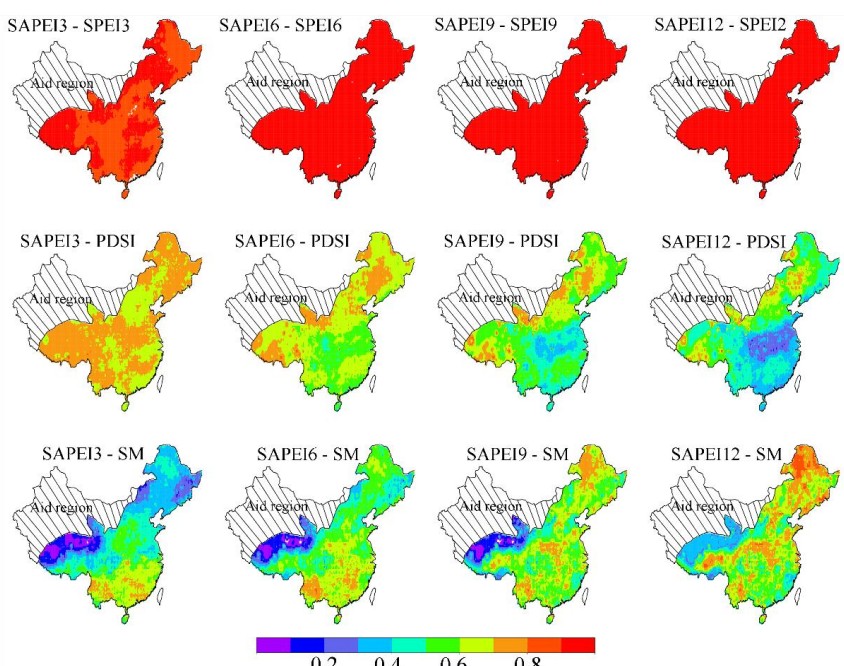


Figure 2 The spatial pattern of the correlations between monthly SAPEI and SPEI/PDSI,
and between daily SAPEI and soil moisture (SM). The monthly SAPEI is computed by
averaging the daily values in each month.

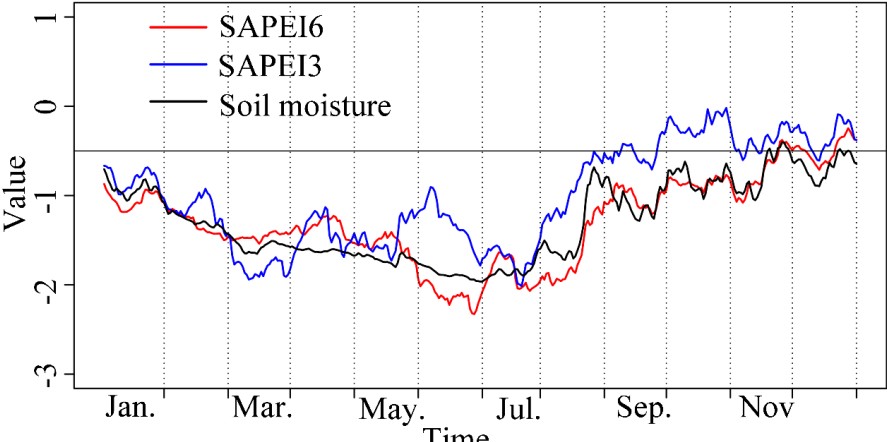


Figure 3 SAPEI (3- and 6-month) and soil moisture series during the 2009/2010 drought
event over the southwest China. The series were spatially average merged series.



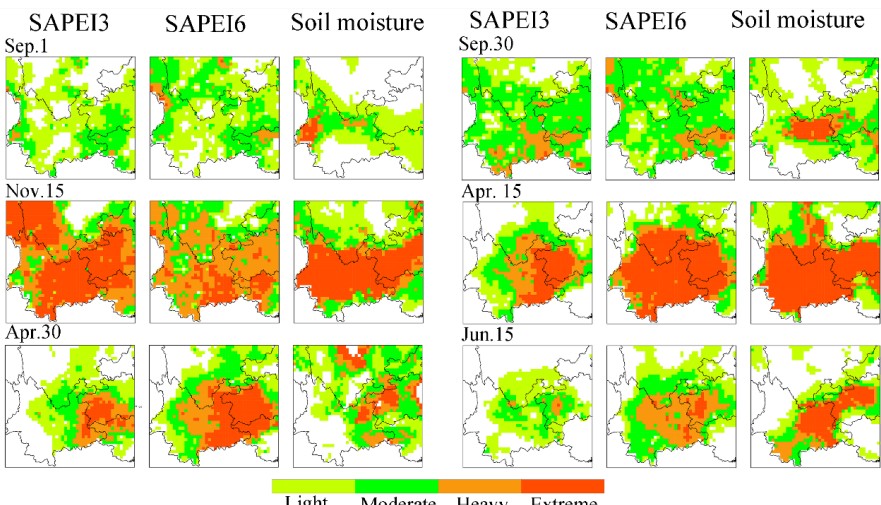


Light    Moderate    Heavy    Extreme

Figure 4 Daily evolutions of the 2009/2010 drought event over the southwest China
monitored by 3- and 6-month SAPEI and soil moisture.

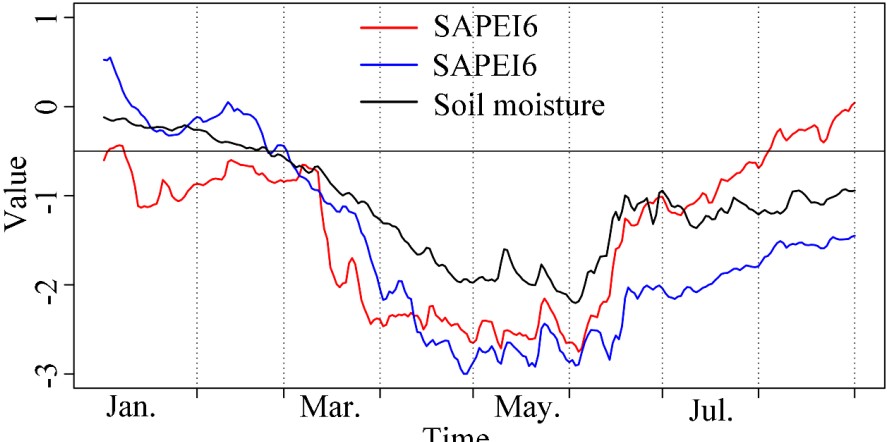


Figure 5 SAPEI (3- and 6-month) and soil moisture series during the 2011 drought
event over the middle and lower reaches of the Yangtze River. The series were spatially
average merged series.






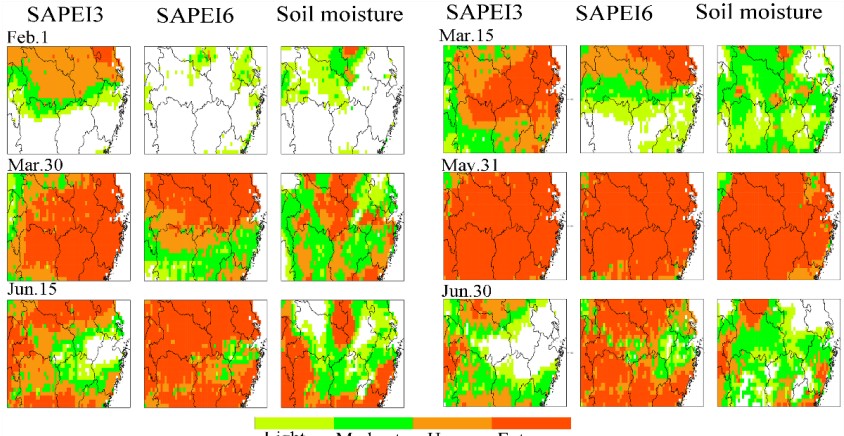

Figure 6 Daily evolutions of the 2011 drought event over the middle and lower reaches

of the Yangtze River monitored by 3- and 6-month SAPEI and soil moisture.



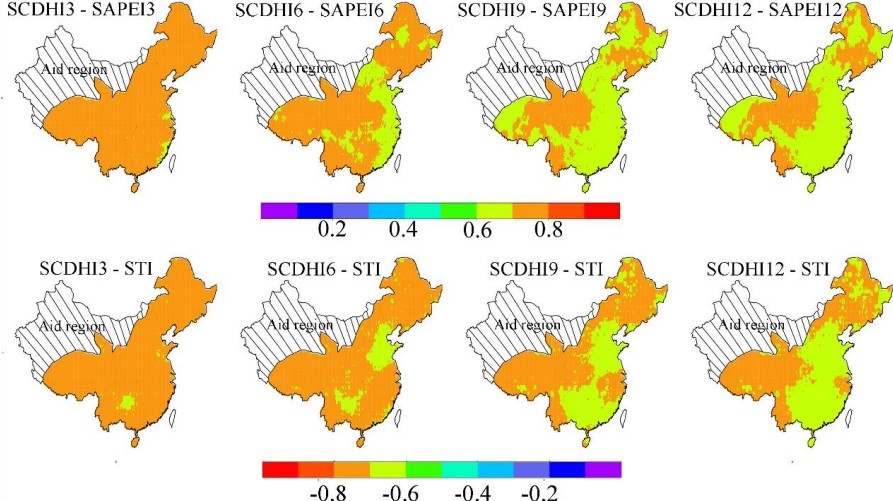

Figure 7 The spatial pattern of the correlations between SCDHI and SAPEI/STI at daily

scale from 1961 to 2018.





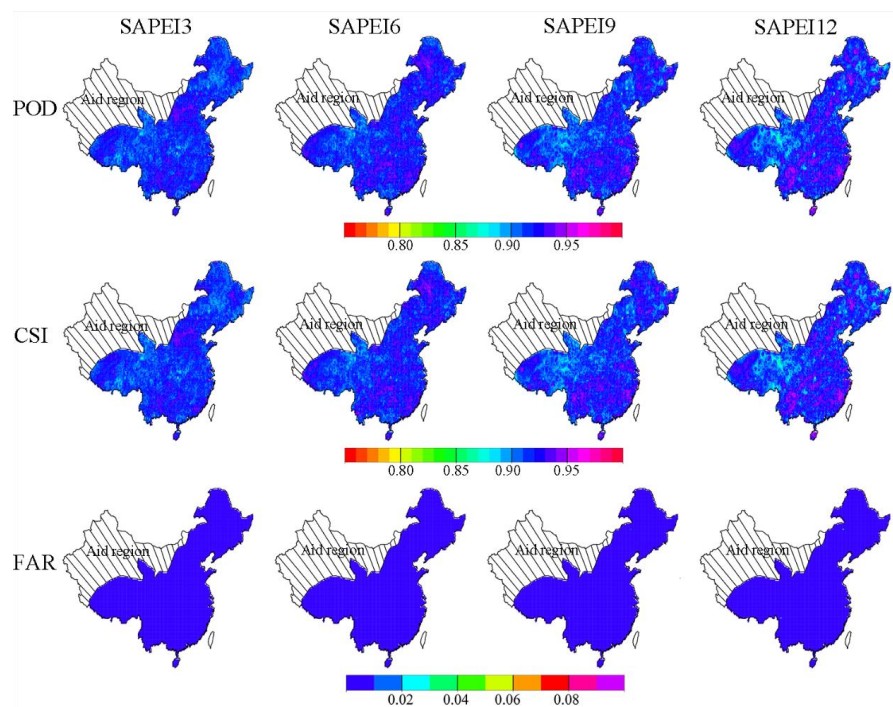


Figure 8 The spatial pattern of POD, FAR, and CSI for SCDHI at 3-, 6-, 9- and 12-
month time scales from 1961 to 2018.





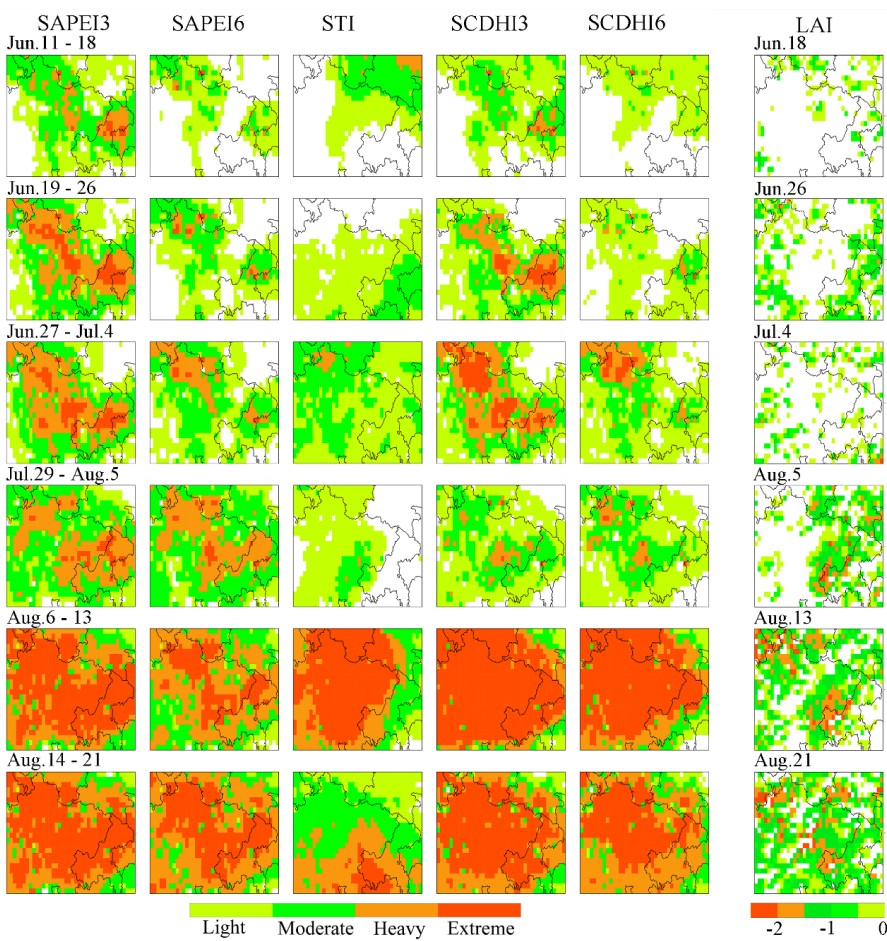


Figure 9. The spatial evolutions of the compound dry and hot event over the Sichuan-

Chongqing region in 2006 and its impact on vegetation.







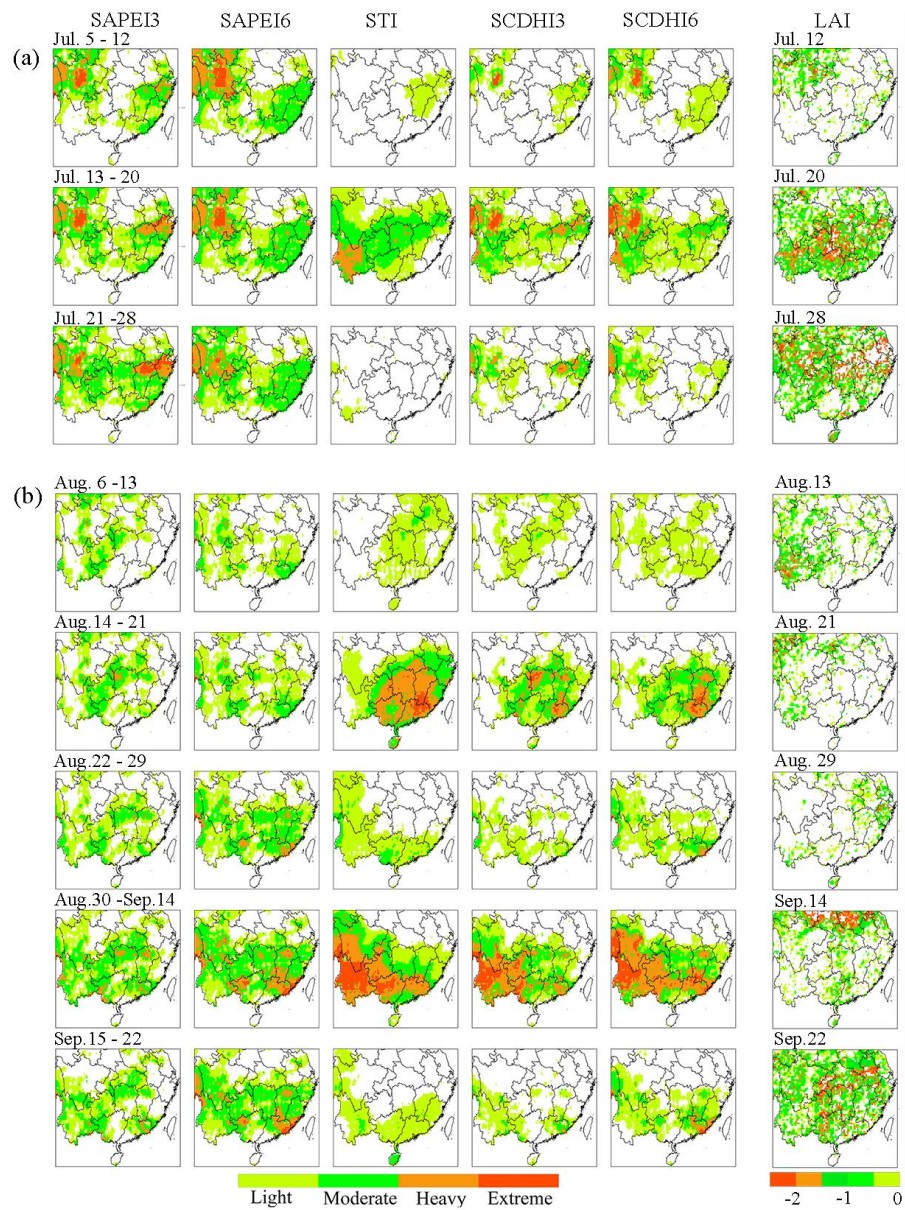


Figure 10 The spatial evolutions of the compound dry and hot event over the southern
China in 2009 and its impact on vegetation.



ЯЯ


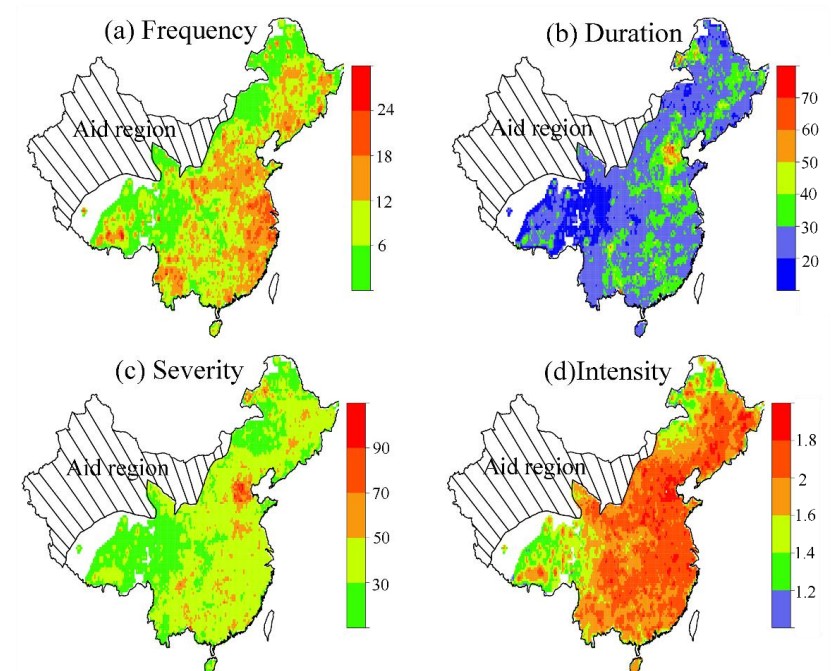


Figure 11 The spatial pattern of the characteristics of the compound dry and hot event
in China from 1961 to 2018.








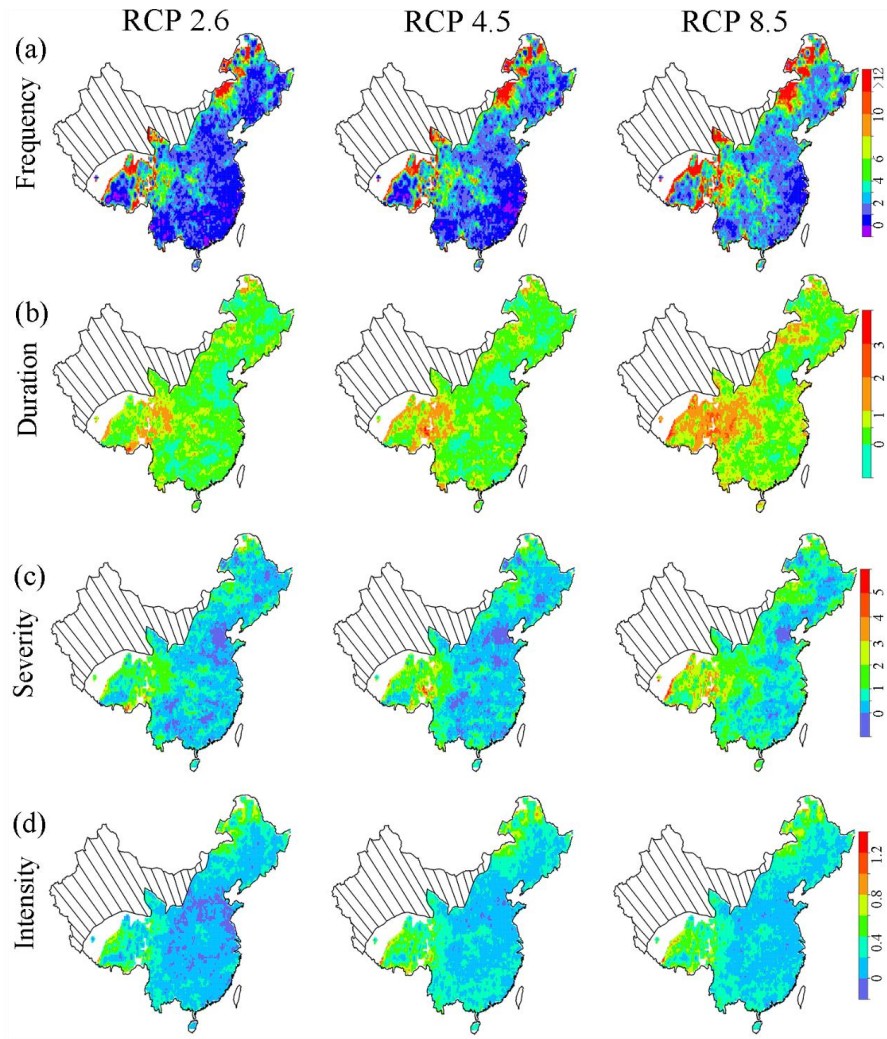


Figure 12 Future changes in characteristics of the compound dry and hot events under
the RCP 2.6, RCP4.5 and RCP8.5 scenarios. The change values were the ratio of the
future value to the reference values. Reference period: 1961-2018, and future period:

903  2050-2100.







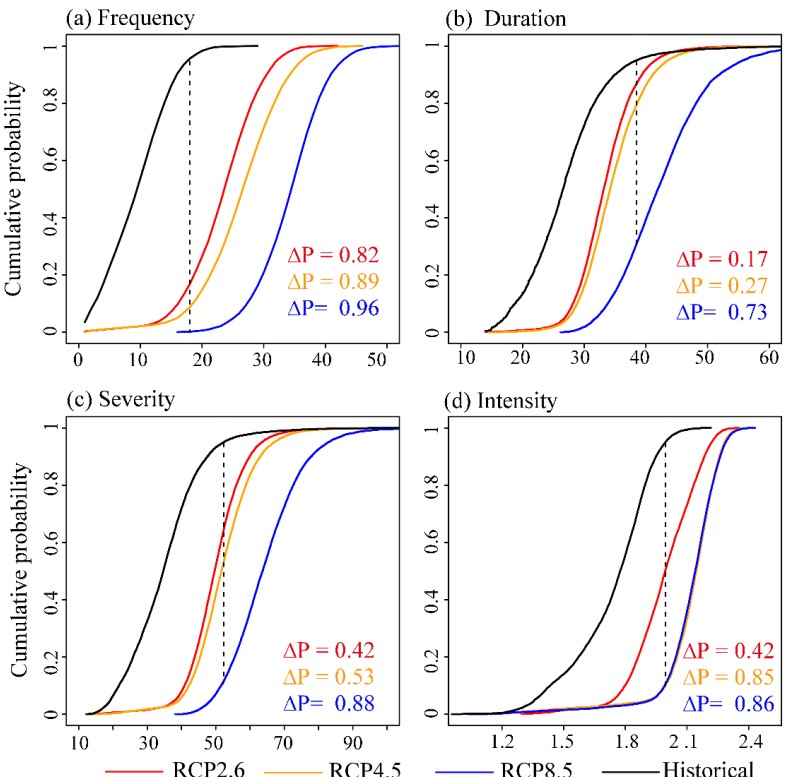

Figure 13 Cumulative probability functions of characteristics of the compound drought

and hot events in historical and future period. The vertical lines denote the probability

of the 95[th] percentile value during the historical period. $\Delta P$ denotes the changes in the

probability of the 95[th] percentile value between the historical period and the future

period. Reference period: 1961-2018, and future period: 2050-2100. The red, orange,

and blue fonts refer to the change values under RCP 2.6, 4.5 and 8.5 scenario,

respectively.