# Peer review of "A standardized index for assessing sub-monthly compound"

_Hydrology and Earth System Sciences, 2020_

## Referee Comment (RC1) · Anonymous Referee #1 · 16 Aug 2020

Understanding the compound dry and hot events is very important to human being society and environments. This study proposes a new compound drought and heat index on daily scale, SCDHI, based on SAPEI and STI. This index is useful to quantify sub-monthly characteristics of compound dry and hot events. The topic is very interesting and suitable for HESS. I recommend the manuscript for acceptance with a minor revision. The detailed comments are provided below:

1) This study focuses the non-arid areas in China. Is SCDHI suitable for the arid areas?

2) There was a similar index for characterizing CDHEs (Hao et al., 2020). I suggest the authors to discuss the difference between this study and the study of Hao et al. (2020), and highlight the novelty of this study in the Introduction section.

[Figure]

Hao, Z., Hao, F., Singh, V. P., Ouyang, W., Zhang, X., & Zhang, S. (2020). A joint extreme index for compound droughts and hot extremes. Theoretical and Applied Climatology, 1-8.

3) Why is the growing season selected to identify CDHEs in Section 3.3? Please explain a little bit more on it.

4) Abstract: the regional difference exists in the future change of the CDHE characteristics. The authors may want to add this in the abstract.

5) P143: how reliable is interpolated data based on the kriging method? Did the author evaluate the interpolated 0.25-degree data?

6) P152: what is the standard number of GB/T 20481-2017? It would be clearer if the authors add some more information on it.

7) P155: soil moisture data in different depths is available in the GLDAS product. Why did the authors choose the root zone soil moisture to evaluate the drought indices? How about soil moisture in the surface layer and in total column?

8) P163: the resolutions of eight climate models are different. Are the results from these models resampled to the same resolution?

9) P164: five is missing after phase.

10) P448: what does the national weather reports look like? I did not see the information on the two CDHEs from the national weather reports.

11) Figs. 3 and 5: is soil moisture is represented by the standardized anomaly? If yes, please briefly describe this. And what is the solid black line all the way from the beginning time down to the ending time?

12) Figs. 4, 6, and 10: please add the longitude and latitude on the figures.

13) Fig. 8: I cannot see the difference among three panels in the last line. Is it because

an inappropriate colobar is used?

14) Figure 11d): the numbers 1.8 and 2 in the colorbar are placed wrongly. They should be exchanged.

15) Figs. 12 and 13: is the historical period used here 1961-2005 or 1951-2018? The authors mentioned that they obtained the model outputs for the 1961-2005 period in Section 2.1. However, the 1961-2005 period does not show up in the results. And is the historical data from the CMIP5 climate models or from the interpolated observations? If the observational data is used as the reference, how the authors resolve the resolution difference between the observational data and the model results?

16) Please check through the manuscript and correct all the grammar mistakes.

---

## Referee Comment (RC2) · Anonymous Referee #2 · 2 Sep 2020

**HESS-2020-383 review**

The paper discusses a standardized index for assessing compound dry and hot conditions. Overall, I find the paper not in a really good shape, and I have to admit that I found it really hard to read due to the excessive amount of acronyms. The paper is so technical that for a reader who does know something about the topic, it is still very hard to follow. For me it did not became entirely clear what are now the new insights that can be learned by creating this new index that were not known before. I also think that the authors should make a new selection of figures and reduce the paper to the essentials, because with the figures in the text and the supplementary material there are so many panels showing China that it becomes overwhelming to the reader. I put some comments below that could help in improving the paper.

It could be good to mention already in the title that this study only concerns China. The paper does not deliver a universal index for compound dry and hot conditions, but one that is only developed for application in China.

As a reviewer, it did not become completely clear to me what the exact problem is of combined dry and hot conditions. There are many examples, but their explanation does not really get to the core: why do we need an indicator for dry and hot? Please improve this in the revision.

I find the methods a little ill-described. There are many references back to previous papers, but please list the equations of the equations that you take from these papers, because now the reader has to look up essential information in previous papers. Also, please be exact what the source of the input data is that is needed to compute all the variables that you need.

Line 203: how does one use a probability distribution to create daily time series, and against what is it fitted? I do not understand the procedure.

Line 219: what is copula theory?

Lines 226-250: This could use some explanatory figures. It is nearly impossible to understand for a reader that is not familiar with the specialized methods that are used here.

Line 265: I think that there are more appropriate and far older references for the definition of the POD and FAR.

Section 3.1: What is the added value from SAPEI compared to much simpler metrics as soil moisture, or if that is not available P-E, or an simple estimation of evapotranspiration?

There are too many references to the supplementary material throughout the text. I suggest the authors reevaluate the necessity for each of the figures and come up with a set that is crucial to the story. This is not a research letter, there is more than enough

space.

Line 462. If a hot index is based on absolute temperature, it seems trivial that places that are closer to the equator at low altitudes have the largest probability of a hot event. Can you explain more about the location where the outcome surprised you, or where new insights were found?

Lines 485 and further: How are the RCP scenarios computed in your index? This does not seem trivial to me, how is the input acquired? It would be nice to know which of the observed increases in due to temperature alone and which due to more complex interactions?

―――――――――――――――――――

---

## Author Comment (AC1) · 12 Oct 2020

Reviewer: Understanding the compound dry and hot events is very important to human being society and environments. This study proposes a new compound drought and heat index on daily scale, SCDHI, based on SAPEI and STI. This index is useful to quantify sub-monthly characteristics of compound dry and hot events. The topic is very interesting and suitable for HESS. I recommend the manuscript for acceptance with a minor revision. The detailed comments are provided below:

Reviewer (1): This study focuses the non-arid areas in China. Is SCDHI suitable for the arid areas? Author's Reply (1): Thank you for your comment. In this study, we did not assess the application of SCDHI in arid areas in China, for three reasons: 1)

replenishment of water resources in the arid region is mainly from melted glacial or perennial frozen soil, not from precipitation. The statistical drought indices are usually limited in revealing drought in such complex situation; 2) meteorological observations in arid regions are too scarce to conduct robust analysis (Wu et al., 2007; Xu et al., 2015); 3) from a practical perspective, calculating drought indices across arid region with large-scale desert regions is less meaningless (Tomasâ ĂŘBurguera et al., 2020). Thus, we did not evaluate the application of SCDHI in arid region. In further study, we will try to develop the compound dry-hot index adopted in arid regions. We will clary this point in data section of the revised manuscript. References: Tomas‐Burguera, M., Vicente‐Serrano, S. M., Peña‐Angulo, D., Domínguez‐Castro, F., Noguera, I., & El Kenawy, A. Global characterization of the varying responses of the Standardized Evapotranspiration Index (SPEI) to atmospheric evaporative demand (AED). Journal of Geophysical Research: Atmospheres, e2020JD033017. Xu, K., Yang, D., Yang, H., Li, Z., Qin, Y., & Shen, Y. (2015). Spatio-temporal variation of drought in China during 1961–2012: A climatic perspective. Journal of Hydrology, 526, 253-264. Wu, H., Svoboda, M. D., Hayes, M. J., Wilhite, D. A., & Wen, F. (2007). Appropriate application of the standardized precipitation index in arid locations and dry seasons. International Journal of Climatology: A Journal of the Royal Meteorological Society, 27(1), 65-79.

Reviewer (2): There was a similar index for characterizing CDHEs (Hao et al., 2020). I suggest the authors to discuss the difference between this study and the study of Hao et al. (2020), and highlight the novelty of this study in the Introduction section. Hao, Z., Hao, F., Singh, V. P., Ouyang, W., Zhang, X., & Zhang, S. (2020). A joint extreme index for compound droughts and hot extremes. Theoretical and Applied Climatology, 1-8. Author's Reply (2): Thank you for your recommendation. The study of Hao et al. (2020) provides a good background for our study and partially inspired the idea to develop SCDHI. We will add the following explanation how the SCDHI differs from that of Hao et al. (2020) in the revised manuscript: "Hao et al. (2019, 2020) recently proposed the standardized compound event indicator and compound dry-hot index to assess the severity of compound dry and hot events by jointing the marginal distribution of standardized precipitation index (SPI) and standardized temperature index (STI) using the copula theory. These two joint indices provide useful tools to improve our understanding of the frequency, spatial extent and severity of the compound dry-hot event. However, the index is inevitably subjected to some shortcomings including the fixed monthly scale and the disregard of evapotranspiration, which may limit its use in monitoring the detailed evolution of compound dry and hot events."

Reviewer (3): Why is the growing season selected to identify CDHEs in Section 3.3? Please explain a little bit more on it. Author's Reply (3): Thank you for your comment and suggestion. The compound dry-hot events were examined during the growing season (April-September) because this is the time when compound dry-hot events cause major impacts on many sectors such as agriculture. Due to the strong seasonal cycle in temperature and precipitation, if focusing on relative exceedance thresholds and mixing seasons, it would be difficult to interpret. We will add this explanation in the revised manuscript.

Reviewer (4): Abstract: the regional difference exists in the future change of the CDHE characteristics. The authors may want to add this in the abstract. Author's Reply (4): Thank you for your suggestion. Indeed, there are differences between regions for future change of the CDHE characteristics. Under RCP 8.5 scenario, CDHE in the central to west parts of China is expected to markedly increase by more than five times; duration in mid-west China potentially increases by approximately 1.5 times; severity over mid-west China is expected to strengthen more than 3 times. We would add the following text in the abstract: "Under the RCP8.5 scenario, the duration, severity, and frequency across mid-west China would increase by at least 1.5 times".

Reviewer (5): P143: how reliable is interpolated data based on the kriging method? Did the author evaluate the interpolated 0.25-degree data? Author's Reply (5): Thank you for your comment. A reliable interpolation method is important to provide fundamental data for research. To generate reliable gridded data in China, previous studies have compared different interpolation methods (e.g., ordinary nearest neighbor, local polynomial, radial basis function, inverse distance weighting, and ordinary kriging), and they found that the ordinary kriging method shows the best performance and yields higher interpolation accuracy than the other methods (Chen et al., 2010; Lin et al., 2002). Datasets based on the kriging method have also been used extensively for drought analyses (Liu et al., 2016; Wu et al., 2013; Shen et al., 2019). Based on these previous research findings, the kriging method was thus used in this study. We will clarify this in data section of the revised manuscript: "The observational station data were interpolated to $0.25 \times 0.25°$ gridded data by kriging method, as it yields higher interpolation accuracy than the other commonly used methods such as ordinary nearest neighbor and inverse distance weighting (Liu et al., 2016)." References: Chen, D., Ou, T., Gong, L., Xu, C. Y., Li, W., Ho, C. H., & Qian, W. (2010). Spatial interpolation of daily precipitation in China: 1951–2005. Advances in Atmospheric Sciences, 27(6), 1221-1232. Lin, Z. H., Mo, X. G., Li, H. X., & Li, H. B. (2002). Comparison of three spatial interpolation methods for climate variables in China. Acta Geographica Sinica, 57(1), 47-56. Liu, Z., Wang, Y., Shao, M., Jia, X., & Li, X. (2016). Spatiotemporal analysis of multiscalar drought characteristics across the Loess Plateau of China. Journal of Hydrology, 534, 281-299. Shen, Z., Zhang, Q., Singh, V. P., Sun, P., Song, C., & Yu, H. (2019). Agricultural drought monitoring across Inner Mongolia, China: Model development, spatiotemporal patterns and impacts. Journal of Hydrology, 571, 793-804. Wu, J., Zhou, L., Liu, M., Zhang, J., Leng, S., & Diao, C. (2013). Establishing and assessing the Integrated Surface Drought Index (ISDI) for agricultural drought monitoring in mid-eastern China. International Journal of Applied Earth Observation and Geoinformation, 23, 397-410.

Reviewer (6): P152: what is the standard number of GB/T 20481-2017? It would be clearer if the authors add some more information on it. Author's reply (6): Thank you for your comment. The PDSI is a semi physical drought index based on land surface water balance. The parameters of the standardized procedure of the conventional PDSI, including the climatic characteristic and duration factors, are empirically derived using the meteorological data of the central USA with its semi-arid climate. Therefore,

the portability and spatial comparability of the conventional PDSI are relatively poor in other regions. To develop a PDSI suited for China, the PDSI calculation procedure was revised based on long-term meteorological data of several in-situ stations distributed over China that represent the climate characteristic of mainland China. A China national standard of classification of meteorological drought with standard number of GB/T 20481-2017 provides the corrected calculation procedure of the PDSI specific for China. We will add the calculation procedure of PDSI of the GB/T 20481-2017 in the supplementary material.

Reviewer (7): P155: soil moisture data in different depths is available in the GLDAS product. Why did the authors choose the root zone soil moisture to evaluate the drought indices? How about soil moisture in the surface layer and in total column? Author's reply (7): Thank you for your comments. Some soil moisture datasets in the GLDAS product provides different depths, e.g., the NOAH model of GLDAS has a total of 4 layers of thickness: 0-10, 10-40, 40-100, and 100-200 cm, while NOAH only has monthly temporal resolution. The CLSM product used in this study does not have explicit vertical levels, instead soil moisture is represented in Surface (0-2cm), and Root Zone (0-100cm). Root zone soil moisture is chosen over the surface soil moisture because it has the applicability of characterizing drought and has lower noise relative to surface soil moisture (Hunt et al., 2009; Osman et al., 2020). For drought monitoring, this product has the advantage of offering spatially and temporally complete root zone soil moisture estimates on a grid. Furthermore, standard drought indices based on a time scale of three months (or longer) seem to be more representative of drought behaviors in deeper soil layers (Fig. 6 in Nicolai-Shaw et al., 2017). We will add the following text in the data section: "The Community Land Model product does not have explicit vertical levels, instead soil moisture is represented for the surface (0-2cm), and the root zone (0-100cm). Root zone soil moisture is chosen over the surface soil moisture due to its applicability to characterize similar droughts as those captured by drought indices with time scales of three months or longer (Nicolai-Shaw et al., 2017); moreover, it has lower noise relative to surface soil moisture (Hunt et al., 2009; Osman et al., 2020)."

References: Hunt, E. D., Hubbard, K. G., Wilhite, D. A., Arkebauer, T. J., & Dutcher, A. L. (2009). The development and evaluation of a soil moisture index. International Journal of Climatology, 29(5), 747-759. Nicolai-Shaw, N., J. Zscheischler, M. Hirschi, L. Gudmundsson, and S. I. Seneviratne (2017). A drought event composite analysis using satellite remote-sensing based soil moisture. Remote Sensing of Environment 203, 216-225. Osman, M., Zaitchik, B. F., Badr, H. S., Christian, J. I., Tadesse, T., Otkin, J. A., & Anderson, M. C. (2020). Flash drought onset over the Contiguous United States: Sensitivity of inventories and trends to quantitative definitions. Hydrology and Earth System Sciences Discussions, 1-21.

Reviewer (8): P163: the resolutions of eight climate models are different. Are the results from these models resampled to the same resolution? Author's reply (8): Thank you for your question. We are sorry that we did not provide a clear description of how the data was processed. Earth system models (ESMs) provide useful information of future climate projections through global-scale simulations. However, the coarse resolution of ESMs restricts their use in many sub-region-scale applications, requiring downscaling of climate model output (Chen et al., 2019; Fenta and Disse, 2018). In this study, the bias-corrected climate imprint method, a statistical downscaling method based on the delta approach, was applied to downscale the climate model output to a spatial resolution of 0.25°. We will clarify this in data section of the revised manuscript: "In this study, the bias-corrected climate imprint method, a statistical downscaling method based on the delta approach, was used to downscale global climate model output to a spatial resolution of 0.25°."

Reviewer (9): P164: five is missing after phase. Author's reply (9): Thank you for your comment. We would correct it in the revised manuscript.

Reviewer (10): P448: what does the national weather reports look like? I did not see the information on the two CDHEs from the national weather reports. Author's reply (10): Thank you for your question. The national weather report is a public service product provided by China Meteorological Administration (http://www.weather.com.cn/).

Specifically, the CDHE in Sichuan-Chongqing region during the summer of 2006 is reported at http://www.weather.com.cn/zt/kpzt, while the other one during July-September of 2009 was recorded in Yearbook of Meteorological Disasters in China 2010. We will rewrite the part to: "Overall, the changes in these two compound dry-hot events based on SCDHI are consistent with the national weather records (http://www.weather.com.cn/zt/kpzt/)and the Yearbook of Meteorological Disasters in China 2010."

Reviewer (11): Figs. 3 and 5: is soil moisture is represented by the standardized anomaly? If yes, please briefly describe this. And what is the solid black line all the way from the beginning time down to the ending time? Author's reply (11): Thank you for your comment. The soil moisture in Figs. 3 and 5 represents the standardized anomaly. To avoid the effect of seasonality, the soil moisture was fitted by Gamma probability distribution, and then was standardized by normal quantile transformation. The value of solid black line is -0.5, indicating the distinction between drought and non-drought according to our definition. We will clarify this in data section of the revised manuscript: "To avoid the effect of seasonality, soil moisture was fitted to a Gamma distribution and then was standardized by normal quantile transformation."

Reviewer (12): Figs. 4, 6, and 10: please add the longitude and latitude on the figures. Author's reply (12): Thank you for your comment. We will add the longitude and latitude on the figures in revised manuscript.

Reviewer (13): Fig. 8: I cannot see the difference among three panels in the last line. Is it because an inappropriate colobar is used? Author's reply (13): Thank you for your comment. We will revise the figure.

Reviewer (14): Figure 11d): the numbers 1.8 and 2 in the colorbar are placed wrongly. They should be exchanged. Author's reply (14): Thank you for your comment. We are sorry for the mistake and will check throughout the manuscript to avoid such similar mistake. We will revise the figure.

Reviewer (15): Figs. 12 and 13: is the historical period used here 1961-2005 or 1951-2018? The authors mentioned that they obtained the model outputs for the 1961-2005 period in Section 2.1. However, the 1961-2005 period does not show up in the results. And is the historical data from the CMIP5 climate models or from the interpolated observations? If the observational data is used as the reference, how the authors resolve the resolution difference between the observational data and the model results? Author's reply (15): Thank you for your comments and question. In Figs. 12 and 13, the historical periods are from 1961 to 2018, and the observational datasets were used. To match the spatial scale, the bias-corrected climate imprint method, was applied to bias correction and downscale the model output to the same resolution in this study. We will clarify these points in data section of the revised manuscript: "We obtained daily climate variables (i.e., precipitation, temperature, relatively humidity, wind speed, and shortwave and longwave radiations) for the future (2050-2100) periods for the three Representative Concentration Pathways (RCPs) including RCP 2.6 (low emission scenario), RCP 4.5 (moderate emission scenario) and RCP 8.5 (high emission scenario)." "In this study, the bias-corrected climate imprint method (Werner and Cannon, 2016), a statistical downscaling method based on the delta approach was applied to downscale the climate model output to a spatial resolution of 0.25°" Reference: Werner, A. T. and Cannon, A. J.: Hydrologic extremes - An intercomparison of multiple gridded statistical downscaling methods, Hydrol. Earth Syst. Sci., doi:10.5194/hess-20-1483-2016, 2016.

Reviewer (16): Please check through the manuscript and correct all the grammar mistakes. Author's reply (16): Thank you. We will check the revision thoroughly to avoid grammar mistakes.

---

## Author Comment (AC2) · 12 Oct 2020

Reviewer: Interesting objective, interesting method, but hard to read. Reviewer (1): (a) the paper discusses a standardized index for assessing compound dry and hot conditions. Overall, I find the paper not in a really good shape, and I have to admit that I found it really hard to read due to the excessive amount of acronyms. The paper is so technical that for a reader who does know something about the topic, it is still very hard to follow. (b) For me it did not became entirely clear what are now the new insights that can be learned by creating this new index that were not known before. (c) I also think that the authors should make a new selection of figures and reduce the paper to the essentials, because with the figures in the text and the supplementary material there are so many panels showing China that it becomes overwhelming to the reader. I put some

comments below that could help in improving the paper. Author's reply (1): Thank you for your comments and suggestions. (a) We are sorry for the excessive number of acronyms. We will reduce the utilization of the abbreviations in the revised manuscript. (b) Much effort has been made to study the compound dry-hot event in recent years. Utilizing thresholds to define the concurrent events, the frequency of compound events has received much attention (Wu et al., 2019; Zhang et al., 2019). However, this approach fails to measure compound event characteristics (e.g., duration, severity, and intensity), and is inconvenient for comparing compound event characteristics through different climates (Wu et al., 2020). Several indices were thus proposed for analyzing the characteristics of the compound events, such as standardized compound event indicator and standardized compound dry-hot index. These indices provide useful tools to understanding compound dry-hot event characteristics. However, they are subject to some shortcomings including the fixed monthly scale and the disregard of evapotranspiration, which may limit their use in monitoring the detailed evolution of compound dry and hot events. In addition, severe concurrent drought and heat can suddenly strike a region within short duration when extreme weather anomalies persist over the same region (Röthlisberger and Martius, 2019; Wang et al., 2016). Concurrent short-term drought and heatwaves can pose great socio-economic risks (Zhang et al., 2019). There is thus a need to have indices capable of monitoring sub-monthly compound dry-hot conditions. Yet up to now there is no index available for incorporating the joint variability of dry and hot conditions at sub-monthly scale. The proposed SCDHI index provides a new tool to monitor and quantify the characteristics of compound dry-hot events at multiple time scale (e.g., daily, weekly and monthly) to provide detailed information on their initiation, development, termination, and trends. We will put more emphasis on the motivation and benefits of this new index in the introduction of the revised manuscript. (c) We agree that the figures in the first-round manuscript need to be reduced. As the results on 3, 6, 9, 12-month scale SAPEI/SCDHI in Figs. 2, 4, 6, 8, 9, 10 are generally similar, we will only show results on 3-month scale SAPEI and SCDHI, and remove the results of other time scales in these Figures. In addition,

we will remove the Figs. 7 and 13 in revised manuscript. The supplementary materials mainly involve the metrics for selecting copula, and assessment of SAPEI/SCDHI ability to monitor monthly drought/compound dry-hot conditions using real-world typical events. These analyses are necessary but not essential, so placing them in the supplementary material without adding manuscript space. We will reduce the text content related to supplementary materials, and subfigures in supplementary materials, but keep essential figures and content to improve the readability of the paper. References: Röthlisberger, M. and Martius, O.: Quantifying the Local Effect of Northern Hemisphere Atmospheric Blocks on the Persistence of Summer Hot and Dry Spells, Geophys. Res. Lett., doi:10.1029/2019GL083745, 2019. Wang, L., Yuan, X., Xie, Z., Wu, P. and Li, Y.: Increasing flash droughts over China during the recent global warming hiatus, Sci. Rep., doi:10.1038/srep30571, 2016. Wu, X., Hao, Z., Hao, F. and Zhang, X.: Variations of compound precipitation and temperature extremes in China during 1961–2014, Sci. Total Environ., 663, 731–737, doi:10.1016/j.scitotenv.2019.01.366, 2019. Wu, X., Hao, Z., Zhang, X., Li, C. and Hao, F.: Evaluation of severity changes of compound dry and hot events in China based on a multivariate multi-index approach, J. Hydrol., 583, 124580, doi:10.1016/j.jhydrol.2020.124580, 2020. Zhang, Y., You, Q., Mao, G., Chen, C. and Ye, Z.: Short-term concurrent drought and heatwave frequency with 1.5 and 2.0 °C global warming in humid subtropical basins: a case study in the Gan River Basin, China, Clim. Dyn., 52(7–8), 4621–4641, doi:10.1007/s00382-018-4398-6, 2019.

Reviewer (2): It could be good to mention already in the title that this study only concerns China. The paper does not deliver a universal index for compound dry and hot conditions, but one that is only developed for application in China. Author's reply (2): Thank you for your comments and suggestions. Because developing a sub-monthly index requires datasets with high temporal resolution (e.g., daily precipitation, maximum air temperature, mean air temperature, minimum air temperature, relatively humidity, wind speed, and sunshine duration), it is difficult to collect all these daily datasets on a global scale. While the index is computed for China base on readily available datasets, the methodology is universal. Moreover, China has vast territory and complex and diverse climates, and during the past decades, it suffers from frequent and severe compound dry-hot events (Wu et al., 2019). So we think it is a favorable, representative study region to demonstrate our proposed index. We will emphasis that this method is currently used in China in title: "A standardized index for assessing submonthly compound dry and hot conditions: application in China" Reference: Wu, X., Hao, Z., Zhang, X., Li, C., Hao, F. (2019). Evaluation of severity changes of compound dry and hot events in China based on a multivariate multi-index approach. Journal of Hydrology, 583, 124580.

Reviewer (3): As a reviewer, it did not become completely clear to me what the exact problem is of combined dry and hot conditions. There are many examples, but their explanation does not really get to the core: why do we need an indicator for dry and hot? Please improve this in the revision. Author's reply (3): Thank you for your comments. Different combinations of dry and hot conditions lead to different types of impacts including crop failure vegetation impacts or wild fires. Slightly hotter conditions may exacerbate impacts originated from dry conditions (Ribeiro et al., 2020). Furthermore, the correlation between hot and dry conditions renders a naive combination of univariate indicators of hot and dry events unsuitable for estimating combined impacts. A combined dry-hot indicator implicitly accounts for the dependence between hot and dry conditions and provides a univariate metric to measure the intensity of combined stress due to heat and drought. For crops it has been shown that such a bivariate indicator can explain crop yield better than typically used linear regression models (Zscheischler et al., 2017). The author's reply (1) provides further motivation for such an index. We will reduce the number of given examples, and re-write the introduction focusing on the above-mentioned points. References: Ribeiro, A. F. S., Russo, A., Gouveia, C. M., Páscoa, P., and Zscheischler, J.: Risk of crop failure due to compound dry and hot extremes estimated with nested copulas, Biogeosciences Discuss., in review, 2020. Zscheischler, J., Orth, R., and Seneviratne, S. I.: Bivariate return periods of temperature and precipitation explain a large fraction of European crop yields, Biogeosciences, 14, 3309–3320, 2017.

Reviewer (4): I find the methods a little ill-described. There are many references back to previous papers, but please list the equations of the equations that you take from these papers, because now the reader has to look up essential information in previous papers. Also, please be exact what the source of the input data is that is needed to compute all the variables that you need. Author's reply (4): Thank you for your comments and suggestions. We are sorry for the unclear description on methods. In this study, only the SAPEI refers to a previous paper, which only involves an equation used to calculate this index, i.e., equation (1). We will add this equation in the text and clarify the input data of the two indices in method section of the revised manuscript: "The SCDHI calculation relies on STI and SAPEI. STI is calculated from maximum temperature, while SAPEI is calculated from precipitation and potential evapotranspiration. The Penman-Monteith method is used to calculate the potential evapotranspiration, requiring maximum air temperature, mean air temperature, minimum air temperature, relatively humidity, wind speed, and sunshine duration."

Reviewer (5): Line 203: how does one use a probability distribution to create daily time series, and against what is it fitted? I do not understand the procedure. Author's reply (5): Thank you for your comment. The probability is not used to create daily time series, but rather, is applied to fit a time series. We will add a case of SAPEI calculation in the supplementary materials: "Taking the calculation of SAPEI on May 1st of each year (1961-2018) as an example, with respect to 3-month scale SAPEI, the total water surplus or deficit in three months before May 1st of each year is calculated to represent the dry and wet conditions on May 1st, and thus, there are 58 values representing the dry and wet conditions on May 1st of each year. The water surplus or deficit was calculated through the difference between precipitation and potential evapotranspiration. For calculating a standardized index, a probability distribution was used to fit the daily time series (58 values), which was then transformed into a standard normal distribution (resulting in SAPEI) using the classical approach of Barton et al. (1965)." Reference: Barton, D. E., Abramovitz, M. and Stegun, I. A.: Handbook of Mathematical Functions with Formulas, Graphs and Mathematical Tables., J. R. Stat.

[Figure]

Soc. Ser. A, doi:10.2307/2343473, 1965.

Reviewer (6): Line 219: what is copula theory? Author's reply (6): Thank you for your comment. Developed by Sklar (1959), copulas are functions that link univariate distribution functions to form multivariate distribution functions. The merit of using copulas to construct multivariate distributions is that copulas can separate the dependence effects from the marginal distribution effects. Construction of multivariate distribution is thus reduced to study the relations among the correlated random variables if marginal distributions are given. For two random variables, Sklar's Theorem states that if $F_{X,Y}$ (x, y) is a two-dimensional distribution function with marginal distributions $F_X(x)$ and $F_Y$ (y), and there exists a copula C such that: $F_{X,Y}$ (x, y) =C ($F_X$ (x), $F_Y$ (y)) Conversely, for any univariate distributions $F_X(x)$ and $F_Y$ (y) and any copula C, the function $F_{X,Y}$ (x, y) is a two-dimensional distribution function with marginal distributions $F_X(x)$ and $F_Y$ (y). Furthermore, if $F_X(x)$ and $F_Y$ (y) are continuous, then C is unique. Under the assumption that the marginal distributions are continuous with probability density functions $f_X$ (x) and $f_Y$ (y), the joint probability density function turns out to be: $f_{X,Y}$ (x, y)=$c(F_X(x), F_Y(y))f_X(x)f_Y(y)$ where c is the density function of C. The books of Nelsen (2006) introduce a copula theory in detail. We will add some introduction of copula theory function in the supplementary materials. Reference: Sklar, K.: Fonctions de repartition a n Dimensions et Leura Marges, Publ. Inst. Stat. Univ. Paris, 8, 229–231, 1959.

Reviewer (7): Lines 226-250: This could use some explanatory figures. It is nearly impossible to understand for a reader that is not familiar with the specialized methods that are used here. Author's reply (7): Thank you for your comment and suggestion. Fig. S4 in the supplementary materials has already illustrated the SCDHI development. We will move Fig. S4 to the main text.

Reviewer (8): Line 265: I think that there are more approprate and far older references for the definition of the POD and FAR. Author's reply (8): Thank you for your suggestion. We would add the reference: "Winston, H.A., Ruthi, L.J.: Evaluation of RADAP II

severe-storm-detection algorithms. Bull. Am. Meteorol. Soc., 67(2), 145-150, 1986."

Reviewer (9): Section 3.1: What is the added value from SAPEI compared to much simpler metrics as soil moisture, or if that is not available P-E, or an simple estimation of evapotranspiration? Author's reply (9): Thank you for your professional comment. Soil moisture is accepted be the most appropriate variable for agriculture drought monitoring and analyses (Mishra and Singh, 2010). However, there are still lack of long-term and large-scale observational soil moisture datasets due to insufficient observation stations in many parts of the world, especially for developing and undeveloped countries. Because of this, using observational hydrometeorological datasets, the complex physical process models such as the Community Land Model are widely used to simulate soil moisture. However, running such models requires highly trained personnel not usually available at local agencies. In addition, when the model is used locally, it generally needs to be calibrated and verified by observational soil moisture and other hydrometeorological datasets. This certainly limits the wide use of soil moisture as a drought indicator. An evapotranspiration-based drought index provides a useful tool for drought monitoring and analyses. However, in many regions and operational settings, evapotranspiration is derived from potential evapotranspiration (PET) through parameterizations of soil-water and plant-water availabilities that are of questionable value on operational space and time scales; in such cases PET may serve as an independent drought indicator (Hobbins et al., 2016). Recently, the evaporative demand drought index (EDDI) based solely on the PET is used to analyze and monitor flash droughts (McEvoy et al., 2016). However, EDDI only considers for PET and thus is inappropriate for regions with non-constraining soil moisture conditions, e.g., humid regions, given that positive PET anomaly is not representative of actual drought conditions (Vicente-Serrano et al., 2018). The SAPEI relies on the precipitation and potential evapotranspiration. There are generally available observational precipitation and datasets for calculating potential evapotranspiration in most countries around the world. Therefore, SAPEI can be directly calculated using observed, easily accessed meteorological datasets, and the calculation process is simple. In addition, it has multiple time

scales, and the long-time scale SAPEI is sensitive to soil moisture variation. The time scale over which water deficits accumulate becomes very important, and it functionally separates hydrological, agricultural, and other droughts. Drought indices must be associated with a specific time scale to be useful for monitoring and managing different usable water resources (Vicente-Serrano et al., 2010). Overall, the SAPEI meets the requirements of a drought index, given the fact that it shows reliable and robust ability for drought analysis and monitoring. Like SPEI and SPI, SAPEI includes multiple time scales (3-, 6-, 9-, and 12- month) to monitor droughts at monthly resolution. However, SAPEI has the advantage over SPEI regarding sub-monthly drought monitoring. Such an index could help fill the gap between science and applications in that it would be operationally feasible for detecting and monitoring both short-term and long-lasting droughts. We will add some discussion on the added value of SAPEI compared with soil moisture and evapotranspiration-based indices in the revised manuscript. References: Mishra, A. K., Singh, V. P. (2010). A review of drought concepts. Journal of hydrology, 391(1-2), 202-216. Hobbins, M. T., Wood, A., McEvoy, D. J., Huntington, J. L., Morton, C., Anderson, M., Hain, C. (2016). The evaporative demand drought index. Part I: Linking drought evolution to variations in evaporative demand. Journal of Hydrometeorology, 17(6), 1745-1761. McEvoy, D. J., Huntington, J. L., Hobbins, M. T., Wood, A., Morton, C., Anderson, M., Hain, C. (2016). The evaporative demand drought index. Part II: CONUS-wide assessment against common drought indicators. Journal of Hydrometeorology, 17(6), 1763-1779. Vicente-Serrano, S. M., Beguería, S., López-Moreno, J. I., Angulo, M., El Kenawy, A. (2010). A new global 0.5 gridded dataset (1901–2006) of a multiscalar drought index: comparison with current drought index datasets based on the Palmer Drought Severity Index. Journal of Hydrometeorology, 11(4), 1033-1043. Vicente-Serrano, S. M., Miralles, D. G., Domínguez-Castro, F., Azorin-Molina, C., El Kenawy, A., McVicar, T. R., ... Peña-Gallardo, M. (2018). Global assessment of the Standardized Evapotranspiration Deficit Index (SEDI) for drought analysis and monitoring. Journal of Climate, 31(14), 5371-5393.

Reviewer (10): There are too many references to the supplementary material throughout the text. I suggest the authors reevaluate the necessity for each of the figures and come up with a set that is crucial to the story. This is not a research letter, there is more than enough space. Author's reply (10): Thank you for your comments and suggestions. The supplementary materials mainly involve the metrics for selecting copula, and assessment of SAPEI/SCDHI ability to monitor monthly drought/compound dry-hot conditions using real-world typical events. These analyses are necessary but not essential, so placing them in the supplementary material without adding manuscript space is acceptable. If we remove these materials, the ability of the two indices to monitor monthly drought/dry-hot conditions would not be shown completely. We would like to remain them, but select the essential panels and re-write the content related to supplementary materials in the main text.

Reviewer (11): Line 462. If a hot index is based on absolute temperature, it seems trivial that places that are closer to the equator at low altitudes have the largest probability of a hot event. Can you explain more about the location where the outcome surprised you, or where new insights were found? Author's reply (11): Thank you for your comment. STI representing hot condition is calculated by temperature variation within a specific grid point (similar to common drought indices). For example, for a certain grid point, the STI on January 1st are computed based on temperature datasets observed during 1961-2018. In other words, the hot index (STI) is not affected by location and is only related to its changes within the grid point. Hot events are usually defined relative to the local climatology. Fig. 11 shows the characteristics (e.g., frequency) of the compound dry-hot events. Though the event is closely related to the extreme temperature, it reflects the concurrent dry and hot conditions. Extreme absolute temperature may be more frequent at low altitudes, but this does not mean that in such places droughts would occur frequently (as at low altitudes, e.g. equator, plenty precipitation would be expected). In this study, surprisingly, we found that a high frequency of compound events was detected in humid southern China, and the events generally lasted about 25 days (Fig. 11a). Previous studies indicate that the occurrence of extreme climate (e.g. high temperature, low humidity, and sunny skies) can appear within a

short period without resulting in long-lasting compound events, but rather, short-term droughts and heatwave lasting a few weeks (Mo and Lettenmaier, 2015; Zhang et al., 2019). And short-term dry and hot events occur more frequently in southern regions than in other parts of China (Otkin et al., 2018; Wang et al., 2016). South China is a humid region where evapotranspiration is mainly controlled by energy supply because soil moisture is usually sufficient. The evaporation demand could increase significantly during a short period when strong, transient meteorological changes occur. Through influencing evapotranspiration variation, short-term meteorological variables (e.g., so-lar radiation and sunshine duration) are considered important factors driving drought and hot concurrence. For example, large increase in sunshine duration due to clear sky creates excessive evapotranspiration, which in turn decreases soil moisture; more surface sensible heat fluxes are transferred to the near-surface atmosphere to further increase air temperature and makes precipitation rare. These land-atmosphere inter-actions altogether create favorable conditions for concurrent drought and hot events in South China. We will discuss why southern China experience more compound dry-hot events in the revised manuscript based on the description above. References: Mo, K. C., Lettenmaier, D. P. (2015). Heat wave flash droughts in decline. Geophysical Research Letters, 42(8), 2823-2829. Otkin, J. A., Svoboda, M., Hunt, E. D., Ford, T. W., Anderson, M. C., Hain, C., Basara, J. B. (2018). Flash droughts: A review and assessment of the challenges imposed by rapid-onset droughts in the United States. Bulletin of the American Meteorological Society, 99(5), 911-919. Wang, L., Yuan, X., Xie, Z., Wu, P., Li, Y. (2016). Increasing flash droughts over China during the recent global warming hiatus. Scientific reports, 6, 30571. Zhang, Y., You, Q., Mao, G., Chen, C., Ye, Z. (2019). Short-term concurrent drought and heatwave frequency with 1.5 and 2.0 C global warming in humid subtropical basins: a case study in the Gan River Basin, China. Climate dynamics, 52(7-8), 4621-4641.

Reviewer (12): Lines 485 and further: (a) how are the RCP scenarios computed in your index? This does not seem trivial to me, how is the input acquired? (b) It would be nice to know which of the observed increases in due to temperature alone and which due

to more complex interactions? Author's reply (12): Thank you for your comment. (a) To obtain the future climate scenario data, we collect eight global climate models, including CanESM2, CNRM-CM5, CSIRO-Mk3.6, MIROC-ESM, MPI-ESM-LR, BCC-CSM1-1, IPSL-CM5A-LR, and MRI-CGCM3, to project the future climate conditions. These global climate models exhibit good performance to simulate the key features of precipitation and temperature in China. We obtained climate variables (i.e., precipitation, temperature, relatively humidity, wind speed, and shortwave and longwave radiations) for the future periods for the three Representative Concentration Pathways (RCPs) including RCP 2.6 (low emission scenario), RCP 4.5 (moderate emission scenario) and RCP 8.5 (high emission scenario). The bias-corrected climate imprint method, one of the delta statistical downscaling methods, was applied to downscale the climate model output to a same resolution as the observations. Using the downscaling datasets, the SCDHI was computed, and was used to analyze future compound dry-hot events. (b) We will calculate the future SCDHI considering only temperature change, and then this SCDHI will compared to historical reference. Finally, this result will be compared with the Fig. 12 in the first-round manuscript to illustrate future changes in characteristics of the compound dry-hot events due to temperature change.

---

## Referee Report (RR1)

**General comments**

This paper proposed an index "standard compound drought and heat index (SCDHI)" to identify the concurrent dry and hot event. The SCDHI is a combination of two indexes: drought index SAPEI and hot index STI. The advantage of SCDHI is to reflect the dry and hot condition at sub-monthly scale. Such a feature benefits from the use of the SAPEI index which is a daily drought index and enables the drought index calculation at different monthly time scales (e.g., 3, 6, 9, and 12 months). The authors validated the SCDHI index through three evaluation metrics and applied the SCDHI index to three future climate projections. The paper addresses a water resources management question which is within the scope of HESS. However, the innovation of this paper is unclear; the descriptions of the index and experiments are incomplete and not well organized; and a few method choices are not properly justified. Moreover, the paper needs more proofreading. The current presentation is far from the publication criteria of HESS.

**Major comments to the authors**

1. The innovation of this paper is unclear.
    1) The development of SCDHI includes two steps. The first step is developing SAPEI, and the second step is merging SAPEI and STI into SCDHI. If I understand correctly, the development of SAPEI has been published in Li et al. (2020, JHM), and the development of SCDHI looks the same as Hao et al. (2019, JH). The innovation of this work to me is that the authors applied their previously developed SAPEI index to SCDHI. In this regard, I think the novelty of this work is limited and it is not worth publishing in HESS. If this is incorrect, I hope to see the authors explain their novelty compared with Li et al. (2020, JHM) and Hao et al. (2019, JH).
    2) Given that this paper focuses on developing SCDHI based on SAPEI and the development of SAPEI have been published in Li et al. (2020, JHM), I suggest the authors merging the Sections 2.2.1 and 2.2.2 into one section, and adding details about the calculation of the STI index. In this way, the calculation of SCDHI is more complete.
    3) Moreover, to make the calculation of SCDHI look clear, I suggest the authors moving the graphical illustration of the SCDHI construction (Figure S4) from supplementary to the main context.
    4) Considering the focus of this paper, I suggest the authors removing Section 3.1 Evaluation of SAPEI because the development process of SAPEI has been published in Li et al. (2020, JHM). The authors can cite the relevant reference to prove the validation of SAPEI, which will make this paper concise and focused.

2. I'm not satisfied with the structure and descriptions of Section 2 Methods.

1) Regarding PDSI and SPEI (Section 2.1 Data), the introduction of PDSI and SPEI should not be mixed with data (e.g., meteorological dataset), the authors can add a new sub-section to introduce the both indexes. Moreover, please clarify what are the two indexes compared with, SCDHI or SAPEI (lines 136-138)? Last, please explain in detail how to calculate PDSI and SPEI. Currently, there are no explanations for the variables of Equations S1-4, and there are no explanations for the calculation of SPEI.

2) Regarding SCDHI (Section 2.2.2 Construction of SCDHI), how are the marginal distribution and the joint probability distribution calculated (equation (2))? Please add equations or references. Moreover, as I mentioned earlier, please add the calculation equation for STI. Last but not the least, please explain/justify in detail why the Frank copula was chosen based on the three goodness-of-fit measures (lines 260-263).

3) Regarding the three verification metrics (Section 2.2.2 Construction of SCDHI), I think these metrics can be separate from the construction of SCDHI, after all, the three metrics are not part of the SCDHI index but part of the evaluation of SCDHI. Moreover, the authors need to justify why the three verification metrics are selected, for instance, what can each of them reveal, what is the relationship between them, etc.

4) For ease of understanding, I suggest the authors adding a paragraph or sub-section describing the experiment of this work. For instance, after introducing all the data, developed indexes and evaluation metrics, the authors can explain the work flow of this paper, such as firstly validating the developed SCDHI index, and then applying SCDHI to three future climate scenarios. Now the structure of this paper is confusing, only after I read the results section, I understood what the authors want to do.

5) Section 2 Methods should also explain how are the frequency, duration, severity and intensity of the compound dry-hot events defined and calculated (Figs 10, 11). These metrics are used in Section 3.3 Application without proper explanations.

3. Presentation of the paper. The writing of this paper needs to be largely improved. I found it hard to understand some sentences, the definitions of several terms are missing, and the figures are not well titled and explained. Please see the below for details.

**Minor comments to the authors**

1. Please define the term "compound dry-hot event" and give some examples. This term plays a key role in your work, but it lacks an appropriate definition (lines 43-45).
2. Lines 30-31, I'm not sure the purpose of this sentence here.
3. Line 171, the reference "rlilpl" looks not being used in this article.
4. Line 237, "less than" should be "less than or equal to".
5. Lines 331-333, I doubt that the first half sentence is supported by Fig. S6.
6. Some references in writing are unclear. For instance,
1) Lines 57, "this approach".
2) Lines 242-244, "p to the given marginal sets".

3) Line 354, "It".
4) Line 379, "detect" what?
5) Line 455, "they".
7. Many sentences have grammatical errors or are unclear. Please correct them and add elaborations. Below are just some examples.
1) Line 131, change "perennial frozen soil" to "perennially frozen soil".
2) Line 132, change "Chinese arid regions" to "the arid regions of China".
3) Line 135, I guess "less meaningless" should be "less meaningful".
4) Lines 154-156, two "well".
5) Lines 161-165.
6) Lines 225-228
7) Lines 252-253. The citation format is incorrect.
8) Lines 269-274. "is" vs. "are".
9) Lines 284-286.
10) Lines 320-322.
11) Lines 371-379.
12) Lines 403-405.
13) Line 417, change "twenty thousand km2" to "20,000km2".
14) Line 420, change "first case studies" to "the first case study".
15) Lines 454-457.
16) Line 467, past tense.
17) Line 472, please elaborate on the "run theory".
18) Lines 510-516. Please re-write these sentences to make them clear.
19) Figure 12 title has grammar error.
20) Lines 536-538.
21) Lines 543-544.
22) Lines 570-571.
8. Please add references for the following sentences:
1) Lines 146-148.
2) Lines 157.
9. The section numbering of 3.2.1 and 3.2.2 has problems.

---

## Author Response (AR2)

**Reply to Reviewer 1**

Reviewer: Understanding the compound dry and hot events is very important to human being society and environments. This study proposes a new compound drought and heat index on daily scale, SCDHI, based on SAPEI and STI. This index is useful to quantify sub-monthly characteristics of compound dry and hot events. The topic is very interesting and suitable for HESS. I recommend the manuscript for acceptance with a minor revision. The detailed comments are provided below:

Author's reply: We highly appreciate the constructive comments and suggestions.

Reviewer (1): This study focuses the non-arid areas in China. Is SCDHI suitable for the arid areas?

Author's Reply (1): Thank you for your comment. In this study, we did not assess the application of SCDHI in arid areas in China, because of three reasons: (1) replenishment of water resources in the Chinese arid region is mainly from melted glacial or perennial frozen soil, and not from precipitation. The statistical drought indices are usually limited its role in revealing drought in such complex situation. (2) meteorological observations in Chinese arid regions are too scarce to conduct robust analysis (Wu et al., 2007; Xu et al., 2015). (3) From a practical perspective, calculating climate extreme indices across arid region with large-scale desert regions is less meangless (Tomas-Burguera et al., 2020). Thus, we did not evaluate the application of SCDHI in arid region. In further study, we will try to develop the compound dry-hot index adopted arid regions.

We have added explanation in Lines 156-163.

Reference:

Tomas-Burguera, M., Vicente-Serrano, S. M., Peña-Angulo, D., Domínguez-Castro, F., Noguera, I., & El Kenawy, A. Global characterization of the varying responses of the Standardized Evapotranspiration Index (SPEI) to atmospheric evaporative demand (AED). Journal of Geophysical Research: Atmospheres, e2020JD033017.

Xu, K., Yang, D., Yang, H., Li, Z., Qin, Y., & Shen, Y. (2015). Spatio-temporal variation of drought in China during 1961–2012: A climatic perspective. Journal of Hydrology, 526, 253-264.

Wu, H., Svoboda, M. D., Hayes, M. J., Wilhite, D. A., & Wen, F. (2007). Appropriate application of the standardized precipitation index in arid locations and dry seasons. International Journal of Climatology: A Journal of the Royal Meteorological Society, 27(1), 65-79.

Reviewer (2): There was a similar index for characterizing CDHEs (Hao et al., 2020). I suggest the authors to discuss the difference between this study and the study of Hao et al. (2020), and highlight the novelty of this study in the Introduction section. Hao, Z., Hao, F., Singh, V. P., Ouyang, W., Zhang, X., & Zhang, S. (2020). A joint extreme index for compound droughts and hot extremes. Theoretical and Applied Climatology, 1-8.

Author's Reply (2): Thank you for your recommendation. The study of Hao et al. (2020) provides a good background for our study and partially inspired the idea to develop SCDHI.

We have added discussion in Lines 75-85.

Reviewer (3): Why is the growing season selected to identify CDHEs in Section 3.3? Please explain a little bit more on it.

Author's Reply (3): Thank you for your comment and suggestion. The compound dry-hot events were examined during the approximate growing season (April-September) because this is the time where they can cause major impacts. Duo to the strong seasonal cycle in temperature and precipitation and focusing on relative exceedance thresholds, mixing seasons could result in results that are difficult to interpret.

We have added explanation in Lines 528-529.

Reviewer (4): Abstract: the regional difference exists in the future change of the CDHE characteristics. The authors may want to add this in the abstract.

Author's Reply (4): Thank you for your suggestion. Indeed, there are difference between region for future change of the CDHE characteristics. Specifically, under RCP 8.5 scenario, CDHE in the central to west parts of China is expected to markedly increase by more than five times; duration in mid-west China potentially increases by approximately 1.5 times; severity over mid-west China is expected to more than triple under RCP 8.5.

We have added these contents in Abstract. Please see Lines 36-38.

Reviewer (5): P143: how reliable is interpolated data based on the kriging method? Did the author evaluate the interpolated 0.25-degree data?

Author's Reply (5): Thank you for your questions. A reliable interpolation method is important to provide fundamental data for research. To generate reliable gridded data in China, previous studies have compared different interpolation methods (e.g., ordinary nearest neighbor, local polynomial, radial basis function, inverse distance weighting, and ordinary kriging), and they found that the ordinary kriging method shows the best performance and yields higher interpolation accuracy than the other methods (Chen et al., 2010; Lin et al., 2002).

Datasets based on the kriging method have also been used extensively for drought analyses (Liu et al., 2016; Wu et al., 2013; Shen et al., 2019). Based on these previous research findings, the kriging method was thus used in this study, and we did not evaluate the kriging method but rely on previous findings.

We have added explanation in Lines 153-156.

Reference:

Chen, D., Ou, T., Gong, L., Xu, C. Y., Li, W., Ho, C. H., & Qian, W. (2010). Spatial interpolation of daily precipitation in China: 1951–2005. Advances in Atmospheric Sciences, 27(6), 1221-1232.

Lin, Z. H., Mo, X. G., Li, H. X., & Li, H. B. (2002). Comparison of three spatial interpolation methods for climate variables in China. Acta Geographica Sinica, 57(1), 47-56.

Liu, Z., Wang, Y., Shao, M., Jia, X., & Li, X. (2016). Spatiotemporal analysis of multiscalar drought characteristics across the Loess Plateau of China. Journal of Hydrology, 534, 281-299.

Shen, Z., Zhang, Q., Singh, V. P., Sun, P., Song, C., & Yu, H. (2019). Agricultural drought monitoring across Inner Mongolia, China: Model development, spatiotemporal patterns and impacts. Journal of Hydrology, 571, 793-804.

Wu, J., Zhou, L., Liu, M., Zhang, J., Leng, S., & Diao, C. (2013). Establishing and assessing the Integrated Surface Drought Index (ISDI) for agricultural drought monitoring in mid-eastern China. International Journal of Applied Earth Observation and Geoinformation, 23, 397-410.

Reviewer (6): P152: what is the standard number of GB/T 20481-2017? It would be clearer if the authors add some more information on it.

Author's reply (6): Thank you for your question and suggestion. The PDSI is a semi physical drought index based on the land surface water balance. The parameters of the standardized procedure of the conventional PDSI, including the climatic characteristic and duration factors, are empirically derived using the meteorological data of the central USA with its semi-arid climate. Therefore, the portability and spatial comparability of the conventional PDSI are relatively poor in other regions of the world. To develop a PDSI suited for China, the PDSI calculation procedure was revised based on long-term meteorological data of several in-situ stations distributed around China that represent the climate characteristic of mainland China. A China national standard of classification of meteorological drought with standard number of GB/T 20481–2017 provides the corrected calculation procedure of the PDSI specific for China:

$$Z_i = K_m d_i \tag{1}$$

$$K_m = \left( \frac{16.84}{\sum\limits_{j=1}^{12} \overline{D_j} K'_m} \right) K'_m \tag{2}$$

$$K'_m = 1.6 \log_{10} \left[ \left( \frac{\overline{PE_m} + \overline{R_m} + \overline{RO_m}}{\overline{P_m} + \overline{L_m}} + 2.8 \right) / \overline{D_m} \right] + 0.4 \tag{3}$$

$$X_i = 0.755 X_{i-i} + Z_i / 1.63 \tag{4}$$

We have added the calculation procedure of PDSI of the GB/T 20481–2017 in supplementary material.

Reviewer (7): P155: soil moisture data in different depths is available in the GLDAS product. Why did the authors choose the root zone soil moisture to evaluate the drought indices? How about soil moisture in the surface layer and in total column?

Author's reply (7): Thank you for your questions and comments. Some soil moisture datasets in the GLDAS product provides different depths. For instance, the NOAH model of GLDAS has total of 4 layers thickness: 0-10, 10-40, 40-100, and 100-200 cm, while NOAH only has monthly temporal resolution. The CLSM product used in this study does not have explicit vertical levels, instead soil moisture is represented in Surface (0-2cm), and Root Zone (0-100cm). Root zone soil moisture is chosen over the surface soil moisture on account of its appositeness to characterize drought and low noise relative to surface soil moisture (Hunt et al., 2009; Osman et al., 2020). For drought monitoring, this product has the advantage of offering spatially and temporally complete root zone soil moisture estimates on a grid. Furthermore, standard drought indices based on a time scale of three months (or higher) seem to more representative of drought behavior in deeper soil layers (Fig. 6 in Nicolai-Shaw et al., 2017).

We have added illustration in Lines 177-181.

Reviewer (16): Please check through the manuscript and correct all the grammar mistakes.

Author's reply (16): Thank you. We have checked the revision thoroughly for grammar mistakes.

**Reply to Reviewer 2**

Reviewer: Interesting objective, interesting method, but hard to read.

Author's reply: Thank you for your comments and suggestions.

Reviewer (1): (a) The paper discusses a standardized index for assessing compound dry and hot conditions. Overall, I find the paper not in a really good shape, and I have to admit that I found it really hard to read due to the excessive amount of acronyms. The paper is so technical that for a reader who does know something about the topic, it is still very hard to follow. (b) For me it did not became entirely clear what are now the new insights that can be learned by creating this new index that were not known before. (c) I also think that the authors should make a new selection of figures and reduce the paper to the essentials, because with the figures in the text and the supplementary material there are so many panels showing China that it becomes overwhelming to the reader. I put some comments below that could help in improving the paper.

Author's reply: Thank you for your comments and suggestions.

(a) We are sorry for the excessive number of acronyms. We have strongly reduced the utilization of the abbreviations in revised manuscript.

(b) Please allow us to clarify the new insights of this index:

Much effort has been made to study the compound dry-hot event in recent years. Utilizing thresholds to define the concurrent events, the frequency of compound events has received much attention (Wu et al., 2019; Zhang et al., 2019). However, this approach fails to measure compound event characteristics (e.g., duration, severity, and intensity), and is inconvenient for comparing compound event characteristics through different climates (Wu et al., 2020). Several indices were thus proposed for analyzing the characteristics of the compound events, such as standardized compound event indicator and standardized compound dry-hot index. These indices provide useful tools to understanding compound dry-hot event characteristics. However, they are subject to some shortcomings including the fixed monthly scale and the disregard of evapotranspiration, which may limit their use in monitoring the detailed evolution of compound dry and hot events.

In addition, severe concurrent drought and heat can suddenly strike a region within short duration when extreme weather anomalies persist over the same region (Röthlisberger and Martius, 2019; Wang et al., 2016). Concurrent short-term drought and heatwaves can pose great socio-economic risks (Zhang et al., 2019). There is thus a need to have readily available indices capable of monitoring sub-monthly compound dry-hot conditions. A suite of indices have been proposed for the assessments of droughts and heatwaves separately, yet there is no index available for incorporating the joint variability of dry and hot conditions at sub-monthly scale.

The proposed SCDHI index provides a new tool to monitor and quantify the characteristics of compound dry-hot events at multiple time scale (e.g., daily, weekly and monthly) to provide detailed information on their initiation, development, termination, and trends.

We have rewritten the motivation and benefits of this new index in Lines 127-142.

(c) We agree that the figures in the first-round manuscript need to be reduced. Specifically, as the results on 3, 6, 9, 12-month scale SAPEI/SCDHI in Figure 2, 4, 6, 8, 9, 10 are generally similar, we have only shown results on 3-month scale SAPEI/ SCDHI, and removed the similar results on other time scales in these Figures. In addition, we have removed the Figure 7 and 13 in revised manuscript.

The supplementary materials mainly involve the metrics for selecting copula, and assessment of SAPEI/SCDHI ability to monitor monthly drought/compound dry-hot conditions using real-world typical events. These analyses are necessary but not essential, so placing them in the supplementary material without adding manuscript space. We have reduced the text content related to supplementary materials, and subfigures in supplementary materials, but kept essential figure and content to ensure the integrity of paper structure.

Reviewer (4): I find the methods a little ill-described. There are many references back to previous papers, but please list the equations of the equations that you take from these papers, because now the reader has to look up essential information in previous papers. Also, please be exact what the source of the input data is that is needed to compute all the variables that you need.

Author's reply (4): Thank you for your comments and suggestions. We are sorry for the unclear description on methods. In this study, only the SAPEI involve the previous papers, and the manuscript have already shown the equations that were used to calculate this index, i.e., equation (1).

The SCDHI calculation relies on STI and SAPEI. STI is calculated from maximum temperature, while SAPEI is calculated from precipitation and potential evapotranspiration. The Penman-Monteith method is used to calculate the potential evapotranspiration, requiring maximum air temperature, mean air temperature, minimum air temperature, relatively humidity, wind speed, and sunshine duration.

We have added illustration in Lines 214-216.

Reviewer (5): Line 203: how does one use a probability distribution to create daily time series, and against what is it fitted? I do not understand the procedure.

Author's reply (5): Thank you for your question. The probability is not used to create daily time series, but applied to fit a time series.

Please allow us to show a case for SAPEI calculation:

Taking the calculation of SAPEI on May 1 of each year (1961-2018) as an example, with respect to 3-month scale SAPEI, the total water surplus or deficit in three months before May 1 of each year is calculated to represent the dry and wet condition on May 1, and thus, there are 58 values representing the dry and wet conditions on May 1 of each year from 1961 to 2018. The water surplus or deficit was calculated through the difference between precipitation and potential evapotranspiration. For calculating a standardized index, a probability distribution was used to fit the daily time series (58 values), which was then transformed into a standard normal distribution (resulting in the SAPEI) using the classical approach of Barton et al. (1965).

We have added a case of SAPEI calculation in supplementary materials.

Reference:

Barton, D. E., Abramovitz, M. and Stegun, I. A.: Handbook of Mathematical Functions with Formulas, Graphs and Mathematical Tables., J. R. Stat. Soc. Ser. A, doi:10.2307/2343473, 1965.

Reviewer (6): Line 219: what is copula theory?

Author's reply (6): Thank you for your question.

Developed by Sklar (1959), copulas are functions that link univariate distribution functions to form multivariate distribution functions. The merit of using copulas to construct multivariate distributions is that copulas can separate the dependence effects from the marginal distribution effects. Construction of multivariate distribution is thus reduced to studying the relations among the correlated random variables if marginal distributions are given.

Considering a situation with two random variables, Sklar's Theorem states that if $F_{X,Y}(x, y)$ is a two-dimensional distribution function with marginal distributions $F_X(x)$

and $F_Y(y)$, then there exists a copula $C$ such that:

$$F_{X,Y}(x, y) = C(F_X(x), F_Y(y)) \qquad (1)$$

Conversely, for any univariate distributions $F_X(x)$ and $F_Y(y)$ and any copula $C$, the function $F_{X,Y}(x, y)$ defined above is a two-dimensional distribution function with marginal distributions $F_X(x)$ and $F_Y(y)$. Furthermore, if $F_X(x)$ and $F_Y(y)$ are continuous, then $C$ is unique.

Under the assumption that the marginal distributions are continuous with probability density functions $f_X(x)$ and $f_Y(y)$, the joint probability density function then becomes

$$f_{X,Y}(x, y) = c(F_X(x), F_Y(y)) f_X(x) f_Y(y) \qquad (2)$$

Where $c$ is the density function of $C$.

Books of Nelsen (2006) introduce a copula theory in detail.

We have added a brief introduction of copula theory in supplementary materials.

Reviewer (9): Section 3.1: What is the added value from SAPEI compared to much simpler metrics as soil moisture, or if that is not available P-E, or an simple estimation of evapotranspiration?

Author's reply (9): That is a good question and is basically related to the lack of availability of soil moisture data.

Soil moisture would be the most appropriate variable for agriculture drought monitoring and analyses (Mishra and Singh, 2010). However, there are few long-term and large-scale observational soil moisture datasets due to insufficient observation stations around the world, especially for developing regions, which limits it wide use in drought monitoring and analyses. Thus, using observational hydrometeorological datasets, the complex physical process models, such as the Community Land Model, are widely used to simulate the soil moisture. However, running such models requires highly trained personnel not usually available at local agencies. In addition, when the model is used locally, it generally needs to be calibrated and verified by observational soil moisture and other hydrometeorological datasets. This certainly limits the wide use of soil moisture as a drought indicator.

An evapotranspiration-based drought index provides a useful tool for drought monitoring and analyses. However, in many regions and operational settings, evapotranspiration is derived from potential evapotranspiration (PET) through parameterizations of soil-water and plant-water availabilities that are of questionable value on operational space and time scales: in such cases PET may serve as an independent drought indicator (Hobbins et al., 2016). Recently, the evaporative demand drought index (EDDI) based solely on the PET is used to analyze and monitor flash droughts (McEvoy et al., 2016). However, EDDI only considers for PET and thus is inappropriate for regions with non-constraining soil moisture conditions, e.g. humid regions, given that positive PET anomaly is not representative of actual drought conditions (Vicente-Serrano et al., 2018).

The SAPEI relies on the precipitation and potential evapotranspiration. There are generally available observational precipitation and datasets for calculating potential evapotranspiration in most countries around the world. Therefore, SAPEI can be directly calculated using observed meteorological datasets, and the calculation process is simple. In addition, it has multiple time scales, and the long-time scale SAPEI is sensitive to soil moisture variation. It is commonly accepted that drought is a multiscalar phenomenon. The time scale over which water deficits accumulate becomes extremely important, and it functionally separates hydrological, agricultural, and other droughts. Drought indices must be associated with a specific time scale to be useful for monitoring and managing different usable water resources (Vicente-Serrano et al.,

2010). Overall, the SAPEI meets the requirements of a drought index, given the fact that it shows reliable and robust ability for drought analysis and monitoring. Like the SPEI and SPI, SAPEI includes multiple time scales (3-, 6-, 9-, and 12- month) to monitor droughts at monthly resolution. However, SAPEI has the advantage over SPEI regarding sub-monthly drought monitoring. Such an index could help fill a gap between science and applications in that it would be operationally tractable for detecting and monitoring both short-term and sustained droughts.

We have added discussion on the added value of SAPEI compared with soil moisture indices in Lines 421-432.

Reviewer (10): There are too many references to the supplementary material throughout the text. I suggest the authors reevaluate the necessity for each of the figures and come up with a set that is crucial to the story. This is not a research letter, there is more than enough space.

Author's reply (10): Thank you for your comments and suggestions. The supplementary materials mainly involve the metrics for selecting copula, and assessment of SAPEI/SCDHI ability to monitor monthly drought/compound dry-hot conditions using real-world typical events. These analyses are necessary but not essential, so placing them in the supplementary material without adding manuscript space. If we remove these materials, the ability of the two indices to monitor monthly drought/dry-hot conditions could not be verified.

So, we would like to keep them, but have selected the essential panels and reduced the content related to supplementary materials. Please see Lines 348-361 and 376-387.

Reviewer (11): Line 462. If a hot index is based on absolute temperature, it seems trivial that places that are closer to the equator at low altitudes have the largest probability of a hot event. Can you explain more about the location where the outcome surprised you, or where new insights were found?

Author's reply (11): Thank you for your comment and suggestion.

In this study, the STI representing the hot condition is calculated by the temperature variation within a specific grid point (similar to common drought indices). For example, with respect to one certain grid point, the 1 January STI are computed on the 1 January temperature datasets observed during 1961-2018 at each grid point. In other words, the hot index (STI) is not affected by regional location and are only related to its changes within the grid point. Hot events are always only hot relative to the local climatology.

In addition, the Figure 11 shows the characteristics (e.g., frequency) of the compound dry-hot events. Though the compound dry-hot event is closely related to the extreme temperature, it reflects the concurrent dry and hot conditions. Extreme temperature is more frequent in some regions, but there may be relatively less compound dry-hot events due to less droughts.

In this study, we found that a high frequency of compound events was detected in southern China, and the events generally lasted about 25 days (Fig. 11a). The occurrence of extreme climate (e.g. high temperature, low humidity, and sunny skies) can appear within a short period without resulting in long-lasting compound events, but rather, short-term droughts and heatwave lasting a few weeks (Mo and Lettenmaier, 2015; Zhang et al., 2019). Previous studies state that short-term dry and hot events occur more frequently in southern regions than in other parts of China (Otkin et al., 2018; Wang et al., 2016). South China is a humid region where evapotranspiration is mainly controlled by energy supply because soil moisture is usually sufficient. The evaporation demand could increase significantly during a short period when strong, transient meteorological changes occur. Through influencing evapotranspiration variation, short-term meteorological variables (e.g., solar radiation and sunshine duration) are considered an important factor in drought and hot concurrences. For instance, the largely increase in sunshine duration due to clear sky create excessive evapotranspiration, which in turn decrease soil moisture. More surface sensible heat fluxes are transferred to the near-surface atmosphere to further increase air temperatures and makes precipitation rare These land-atmosphere interactions altogether create favorable conditions for concurrent drought and hot. Therefore, concurrent dry-hot events are likely to occur in south China.

We have added discussion on why southern China experience more compound dry-hot events. Please see Lines 549-562.

Light  Moderate  Heavy  Extreme          -2   -1   0

[Figure]

Figure 98. The spatial evolutions of the compound dry and hot event over the Sichuan-

Chongqing region in 2006 and its impact on vegetation.

[Figure]

SAPEI3    SAPEI6    STI    SCDHI3    SCDHI6    LAI

Light  Moderate  Heavy  Extreme          -2    -1    0

[Figure]

SAPEI3    STI    SCDHI3    LAI

Light    Moderate    Heavy    Extreme

-2    -1    0

Figure  9 The spatial evolutions of the compound dry and hot event over the southern

China in 2009 and its impact on vegetation.

[Figure]

[Figure]

Figure 11 10 The spatial pattern of the characteristics of the compound dry and hot event in China from 1961 to 2018.

[Figure]

Figure 12 11 Future changes in characteristics of the compound dry and hot events under the RCP 2.6, RCP4.5 and RCP8.5 scenarios. The change values were the ratio of the future value to the reference values. Reference period: 1961-2018, and future period:

2050-2100.

[Figure]

Figure 12 Cumulative probability of future changes (multiple) in of the compound dryhot event characteristics. The dash lines indicate future characteristics changes only considered temperature change, while solid lines represent the future changes driven by all variable variation.

[Figure]

Figure 13 Cumulative probability functions of characteristics of the compound drought and hot events in historical and future period. The vertical lines denote the probability of the 95th percentile value during the historical period. ΔP denotes the changes in the probability of the 95th percentile value between the historical period and the future period. Reference period: 1961-2018, and future period: 2050-2100. The red, orange, and blue fonts refer to the change values under RCP 2.6, 4.5 and 8.5 scenario, respectively.

---

## Author Response (AR3)

We highly appreciate the constructive suggestion of the reviewer and the opportunity to submit a revised version of the manuscript. In the revision, we have implemented a number of changes, in particular:

- The explanation of run theory has been moved from the Supplementary Information to the main text.
- The evaluation of the SCDHI based on probability of detection, false alarm ratio and critical success index has been removed, as by definition the SCDHI captures conditions where the STI and SAPEI are extreme at the same time. However, depending on the dependence structure of the STI and SAPEI, the SCDHI events do not coincide to 100% with exceedance of high thresholds in STI and SAPEI (see Figure 2). We thus consider the correlation analyses SCDHI and STI/SAPEI and the evaluation against individual well-known high-impact drought-heat events more informative.
- We removed the future projections based on CMIP5 and instead added a comparison of compound event characteristics between observations and CMIP5 (Fig. 10 and 12 in the revision), complemented by an analysis of the correlations between STI and SAPEI in the CMIP5 models (Fig. S9). In the earlier version of the manuscript, we assessed future changes in CMIP5 compared to present-day observations. However, these changes were primarily due to changes in the copula, because STI and SAPEI where fitted separately to the model output. Further analysis revealed that the CMIP5 model have a strong bias in the dependence between STI and SAPEI, consequently strongly affecting compound event characteristics measured by SCDHI. Here we highlight and discuss these biases, which contributes another novelty to our study.
- We did a thorough language check and updated the colorbars in all maps.

This paper proposed an index "standard compound drought and heat index (SCDHI)" to identify the concurrent dry and hot event. The SCDHI is a combination of two indexes: drought index SAPEI and hot index STI. The advantage of SCDHI is to reflect the dry and hot condition at sub-monthly scale. Such a feature benefits from the use of the SAPEI index which is a daily drought index and enables the drought index calculation at different monthly time scales (e.g., 3, 6, 9, and 12 months). The authors validated the SCDHI index through three evaluation metrics and applied the SCDHI index to three future climate projections. The paper addresses a water resources management question which is within the scope of HESS. However, the innovation of this paper is unclear; the descriptions of the index and experiments are incomplete and not well organized; and a few method choices are not properly justified. Moreover, the paper needs more proofreading. The current presentation is far from the publication criteria of HESS.

Author's reply: We highly appreciate the constructive comments and suggestions. We have revised the manuscript thoroughly and carefully following your comments and suggestions. The line number was added in the manuscript with no changes marked.

**Major comments to the authors**

1. The innovation of this paper is unclear.

1.1 The development of SCDHI includes two steps. The first step is developing SAPEI, and the second step is merging SAPEI and STI into SCDHI. If I understand correctly, the development of SAPEI has been published in Li et al. (2020, JHM), and the development of SCDHI looks the same as Hao et al. (2019, JH). The innovation of this work to me is that the authors applied their previously developed SAPEI index to SCDHI. In this regard, I think the novelty of this work is limited and it is not worth publishing in HESS. If this is incorrect, I hope to see the authors explain their novelty compared with Li et al. (2020, JHM) and Hao et al. (2019, JH).

Author's reply: Thank you for your comment and suggestion.

Please allow us to explain the novelty of this study:

(1) The study of Hao et al. (2019) provides a useful tool for characterizing compound dry-hot events. However, the existing indices only allow for identifying compound dry-hot events at a relatively coarse (i.e., the monthly) temporal resolution (Hao et al., 2019; Wu et al., 2020). Many important characteristics of climate extremes (i.e., drought and heatwaves) are not detectable at monthly scale (Fang et al., 2020; Lu et al., 2014; Otkin et al., 2018). Within the period of a temporal scale, the existing indices use the simple average of temperature/precipitation to indicate the general hot/dry situation of the period (Hao et al., 2019). Nevertheless, the hot extremes usually occur at finer time scales (e.g., days and weeks), and recent research has indicated that severe drought can occur within such short periods as well (e.g., days and weeks) across most of the world, for instance when extreme weather anomalies persist over the same region (so-called flash droughts, Otkin et al., 2018; Zhang et al., 2019). Moreover, the drought index used in existing compound dry-hot indices often relies on precipitation-based drought index (SPI), and leaves out other important drought-related factors (e.g., relative humidity, wind speed, and radiation) (Trenberth et al., 2013; Vicente-Serrano et al., 2010a). To address these limitations, using the SAPEI, the STI and copula theory, we thus developed a compound dry-hot index that can monitor compound dry and hot conditions at multiple time scales (e.g., day, week, and month).

(2) Several studies have been carried out to study compound dry-hot events in China (Chen et al., 2019; Hao et al., 2019; Wu et al., 2020; Zhang et al., 2019; Zhou and Liu, 2018), and these studies help to better understand such events. However, they mostly focused on the frequency and severity of the compound dry-hot event at a relatively coarse (i.e., the monthly) temporal resolution during historical periods without considering their duration and intensity. In addition, the impact of climate model bias on the characteristics of compound dry-hot event in China remain unclear. Understanding climate model biases is a crucial step to assess future compound dry-hot event risk (Villalobos-Herrera et al., 2020). Recent compound dry-hot events have resulted in serious social and economic losses in China, providing a strong motivation for studying compound dry-hot events (Wu et al., 2020; Zhang et al., 2019). Therefore, here we quantify several characteristics of compound dry-hot event and evaluate the impact of climate model biases on its characteristics in China.

(3) Though the SAPEI has been introduced in Li et al. (2020, JHM), the primary limitation of this index is that it has a fixed temporal scale and cannot reflect dry and wet condition at different time scales (i.e., 3-, 6-, 9-, and 12-month). It is however widely accepted that drought is a multiscalar phenomenon. Drought indices must be associated with a specific time scale to be useful for monitoring and managing drought, such as the wide acceptance of the standardized precipitation index (SPI) and standardized precipitation evapotranspiration index (SPEI) (McKee et al., 1993; Vicente-Serrano et al., 2010b). For example, to modify the limitation of the fixed time scale in the Palmer drought severity index (PDSI), a multiple time scale self-calibrated PDSI (denoted as SC-PDSI variant) has been developed for monitoring drought (Liu et al., 2017). Hence, in this study, we modified the SAPEI by developing the multiple time scale (i.e., 3-, 6-, 9-, and 12-month) daily drought index.

Finally, here we evaluate the SCDHI against recently experiences compound dry-hot events and compare its values against the magnitude of vegetation anomalies.

We have illustrated our novelty of this study. Please see Lines 97-108 and 117-130.

Reference:

Chen, L., Chen, X., Cheng, L., Zhou, P., & Liu, Z. (2019). Compound hot droughts over China: Identification, risk patterns and variations. Atmospheric Research, 227, 210-219.

Fang, W., Huang, S., Huang, Q., Huang, G., Wang, H., Leng, G., & Wang, L. (2020). Identifying drought propagation by simultaneously considering linear and nonlinear dependence in the Wei River basin of the Loess Plateau, China. Journal of Hydrology, 591, 125287.

Liu, Y., Zhu, Y., Ren, L., Singh, V. P., Yang, X., & Yuan, F. (2017). A multiscalar Palmer drought severity index. Geophysical Research Letters, 44(13), 6850-6858.

Lu, E., Cai, W., Jiang, Z., Zhang, Q., Zhang, C., Higgins, R. W., & Halpert, M. S. (2014). The day-to-day monitoring of the 2011 severe drought in China. Climate Dynamics, 43(1-2), 1-9.

Hao, Z., Hao, F., Singh, V. P., & Zhang, X. (2019). Statistical prediction of the severity of compound dry-hot events based on El Niño-Southern Oscillation. Journal of Hydrology, 572, 243-250.

McKee, T. B., Doesken, N. J., & Kleist, J. (1993). The relationship of drought frequency and duration to time scales. In Proceedings of the 8th Conference on Applied Climatology (Vol. 17, No. 22, pp. 179-183).

Otkin, J. A., Svoboda, M., Hunt, E. D., Ford, T. W., Anderson, M. C., Hain, C., & Basara, J. B. (2018). Flash droughts: A review and assessment of the challenges imposed by rapid-onset droughts in the United States. Bulletin of the American Meteorological Society, 99(5), 911-919.

Trenberth, K. E., Dai, A., Van Der Schrier, G., Jones, P. D., Barichivich, J., Briffa, K. R., & Sheffield, J. (2014). Global warming and changes in drought. Nature Climate Change, 4(1), 17-22.

Vicente-Serrano, S. M., Beguería, S., López-Moreno, J. I., Angulo, M., & El Kenawy, A. (2010a). A new global 0.5 gridded dataset (1901–2006) of a multiscalar drought index: comparison with current drought index datasets based on the Palmer Drought Severity Index. Journal of Hydrometeorology, 11(4), 1033-1043.

Vicente-Serrano, S. M., Beguería, S., & López-Moreno, J. I. (2010b). A multiscalar drought index sensitive to global warming: the standardized precipitation evapotranspiration index. Journal of climate, 23(7), 1696-1718.

Villalobos-Herrera, R., Bevacqua, E., Ribeiro, A. F., Auld, G., Crocetti, L., Mircheva, B., ... & De Michele, C. (2020). Towards a compound event-oriented climate model evaluation: A decomposition of the underlying biases in multivariate fire and heat stress hazards. Natural Hazards and Earth System Sciences Discussions, 1-31.

Wu, X., Hao, Z., Zhang, X., Li, C., & Hao, F. (2020). Evaluation of severity changes of compound dry and hot events in China based on a multivariate multi-index approach. Journal of Hydrology, 583, 124580.

Zhang, Y., You, Q., Mao, G., Chen, C., & Ye, Z. (2019). Short-term concurrent drought and heatwave frequency with 1.5 and 2.0 °C global warming in humid subtropical basins: a case study in the Gan River Basin, China. Climate dynamics, 52(7-8), 4621-4641.

Zhou, P., & Liu, Z. (2018). Likelihood of concurrent climate extremes and variations over China. Environmental Research Letters, 13(9), 094023.

1.2 Given that this paper focuses on developing SCDHI based on SAPEI and the development of SAPEI have been published in Li et al. (2020, JHM), I suggest the authors merging the Sections 2.2.1 and 2.2.2 into one section, and adding details about the calculation of the STI index. In this way, the calculation of SCDHI is more complete.

Author's reply: We thank the reviewer for pointing this out. We modified the SAPEI introduced in Li et al. (2020), as discussed in our reply 1.1 above. We agree with the reviewer that the calculation of the STI should be described. We have reduced the introduction on SAPEI, added the calculation of the STI, and merged the content on SAPEI and STI into one section. Please see Section 2.2.1 (Lines 188-213).

1.3 Moreover, to make the calculation of SCDHI look clear, I suggest the authors moving the graphical illustration of the SCDHI construction (Figure S4) from supplementary to the main context.

Author's reply: Thank you for the constructive suggestion. We have moved the Figure S4 into the main context. Please see Figure 2 in the revision.

1.4 Considering the focus of this paper, I suggest the authors removing Section 3.1 Evaluation of SAPEI because the development process of SAPEI has been published in Li et al. (2020, JHM). The authors can cite the relevant reference to prove the validation of SAPEI, which will make this paper concise and focused.

Author's reply: Thank you for your comment and suggestion. As mentioned in 1.1, we modified SAPEI in this study, and evaluated the ability of SAPEI to monitor drought at different time scales. We would like to keep Section 3.1 but have reduced the content of Section 3.1 (Lines 292-347).

2. I'm not satisfied with the structure and descriptions of Section 2 Methods.

Author's reply: Thank you for the constructive comments and suggestions, and underlining the clarity of the structure and description of Section 2.

2.1 (1) Regarding PDSI and SPEI (Section 2.1 Data), the introduction of PDSI and SPEI should not be mixed with data (e.g., meteorological dataset), the authors can add a new sub-section to introduce the both indexes. (2) Moreover, please clarify what are the two indexes compared with, SCDHI or SAPEI (lines 136-138)? (3) Last, please explain in detail how to calculate PDSI and SPEI. Currently, there are no explanations for the variables of Equations S1-4, and there are no explanations for the calculation of SPEI.

Author's reply: We appreciate the comment and suggestion.

(1) We have added a new sub-section to introduce both indices (Please see Section 2.3, Lines 263-272).

(2) The two indices were used to compare with SAPEI, reflecting the ability of SAPEI to monitor drought at monthly scale. We have added the clarification in Line 265.

(3) We have explained in detail how to calculate the two indices. Please see the appendix.

2.2 (1) Regarding SCDHI (Section 2.2.2 Construction of SCDHI), how are the marginal distribution and the joint probability distribution calculated (equation (2))? Please add equations or references. (2) Moreover, as I mentioned earlier, please add the calculation equation for STI. (3) Last but not the least, please explain/justify in detail why the Frank copula was chosen based on the three goodness-of-fit measures (lines 260-263).

Author's reply: We appreciate the comment and suggestion.

(1) Since the SAPEI and STI are normally distributed by construction, the normal distribution is used as the marginal distribution. We have clarified this in the text. The Frank copula emerges as the most appropriate copula from a large set of copulas for the task at hand following several evaluation metrics (including AIC and BIC). References have been added in Lines 229-230.

(2) The calculation for STI has been added in Lines 205-213.

(3) A detailed explanation on why we chose the Frank copula has been added in Lines 248-259.

2.3 (1) Regarding the three verification metrics (Section 2.2.2 Construction of SCDHI), I think these metrics can be separate from the construction of SCDHI, after all, the three metrics are not part of the SCDHI index but part of the evaluation of SCDHI. (2) Moreover, the authors need to justify why the three verification metrics are selected, for instance, what can each of them reveal, what is the relationship between them, etc.

Author's reply: As explained in our main statement at the beginning of this document, we have removed the introduction of the three metrics and the analysis based on these three metrics from our study.

2.4 For ease of understanding, I suggest the authors adding a paragraph or sub-section describing the experiment of this work. For instance, after introducing all the data, developed indexes and evaluation metrics, the authors can explain the work flow of this paper, such as firstly validating the developed SCDHI index, and then applying SCDHI to three future climate scenarios. Now the structure of this paper is confusing, only after I read the results section, I understood what the authors want to do.

Author's reply: Thank you for the constructive suggestion. A paragraph to describe the structure of the paper has been added in Lines 131-134.

2.5 Section 2 Methods should also explain how are the frequency, duration, severity and intensity of the compound dry-hot events defined and calculated (Figs 10, 11). These metrics are used in Section 3.3 Application without proper explanations.

Author's reply: Thank you for your comment and suggestion. We have added these explanations when explaining the run theory in detail. This information has been added in Lines 274-289.

3. Presentation of the paper. The writing of this paper needs to be largely improved. I found it hard to understand some sentences, the definitions of several terms are missing, and the figures are not well titled and explained. Please see the below for details.

Author's reply: Thank you for your comment. We have revised the manuscript thoroughly and carefully, including all the comments and suggestions below.

**Minor comments to the authors**

Many thanks to the reviewer for the comments and suggestions.

1. Please define the term "compound dry-hot event" and give some examples. This term plays a key role in your work, but it lacks an appropriate definition (lines 43-45).

Author's reply: Thank you. We simply refer to a concurrent dry and hot conditions as a compound dry-hot event. A definition has been added in Lines 50-51.

2. Lines 30-31, I'm not sure the purpose of this sentence here.

Author's reply: Thank you. We have revised this sentence. Please see Lines 33-35.

3. Line 171, the reference "rlilpl" looks not being used in this article.

Author's reply: Thank you. We have revised this sentence. Please see Lines 178-179.

4. Line 237, "less than" should be "less than or equal to".

Author's reply: Thank you, this has been done. Please see Line 225.

5. Lines 331-333, I doubt that the first half sentence is supported by Fig. S6.

Author's reply: Thank you. We have rephrased this sentence. Please see Lines 334-335.

6. Some references in writing are unclear. For instance,
1) Lines 57, "this approach".
2) Lines 242-244, "p to the given marginal sets".
3) Line 354, "It".
4) Line 379, "detect" what?
5) Line 455, "they".
Author's reply: Thank you. We have revised these sentences.

7. Many sentences have grammatical errors or are unclear. Please correct them and add elaborations. Below are just some examples.
1) Line 131, change "perennial frozen soil" to "perennially frozen soil".
2) Line 132, change "Chinese arid regions" to "the arid regions of China".
3) Line 135, I guess "less meaningless" should be "less meaningful".
4) Lines 154-156, two "well".
5) Lines 161-165.
6) Lines 225-228
7) Lines 252-253. The citation format is incorrect.
8) Lines 269-274. "is" vs. "are".
9) Lines 284-286.
10) Lines 320-322.
11) Lines 371-379.
12) Lines 403-405.
13) Line 417, change "twenty thousand km2" to "20,000km2".
14) Line 420, change "first case studies" to "the first case study".
15) Lines 454-457.
16) Line 467, past tense.
17) Line 472, please elaborate on the "run theory".
18) Lines 510-516. Please re-write these sentences to make them clear.
19) Figure 12 title has grammar error.
20) Lines 536-538.
21) Lines 543-544.
22) Lines 570-571.
Author's reply: Thank you. We revised all sentences accordingly.

8. Please add references for the following sentences:
1) Lines 146-148.
2) Lines 157.
Author's reply: Thank you. We have added appropriate references. Please see Line 154-155/166

9. The section numbering of 3.2.1 and 3.2.2 has problems.
Author's reply: Thank you. We have revised the numbering.